# Vascular surveillance by haptotactic blood platelets in inflammation and infection

Leo Nicolai [1,2], Karin Schiefelbein[1], Silvia Lipsky[1], Alexander Leunig [1], Marie Hoffknecht[1], Kami Pekayvaz [1], Ben Raude[1], Charlotte Marx[1], Andreas Ehrlich[1], Joachim Pircher[1], Zhe Zhang[1], Inas Saleh[1], Anna-Kristina Marel[3], Achim Löf [3], Tobias Petzold[1], Michael Lorenz[1], Konstantin Stark[1], Robert Pick[4,5], Gerhild Rosenberger[1], Ludwig Weckbach[1], Bernd Uhl[6], Sheng Xia[7], Christoph Andreas Reichel[6], Barbara Walzog [4,5], Christian Schulz [1,2], Vanessa Zheden[8], Markus Bender [9], Rong Li [7], Steffen Massberg [1,2✉] & Florian Gaertner [2,8✉]

Breakdown of vascular barriers is a major complication of inflammatory diseases. Anucleate platelets form blood-clots during thrombosis, but also play a crucial role in inflammation. While spatio-temporal dynamics of clot formation are well characterized, the cell-biological mechanisms of platelet recruitment to inflammatory micro-environments remain incompletely understood. Here we identify Arp2/3-dependent lamellipodia formation as a prominent morphological feature of immune-responsive platelets. Platelets use lamellipodia to scan for fibrin(ogen) deposited on the inflamed vasculature and to directionally spread, to polarize and to govern haptotactic migration along gradients of the adhesive ligand. Platelet-specific abrogation of Arp2/3 interferes with haptotactic repositioning of platelets to microlesions, thus impairing vascular sealing and provoking inflammatory microbleeding. During infection, haptotaxis promotes capture of bacteria and prevents hematogenic dissemination, rendering platelets gate-keepers of the inflamed microvasculature. Consequently, these findings identify haptotaxis as a key effector function of immune-responsive platelets.

[1] Medizinische Klinik und Poliklinik I, Klinikum der Ludwig-Maximilians-Universität, 81377 Munich, Germany. [2] DZHK (German Centre for Cardiovascular Research), Partner Site Munich Heart Alliance, 80802 Munich, Germany. [3] Ludwig-Maximilians-Universität, 80799 Munich, Germany. [4] Walter-Brendel-Centre of Experimental Medicine, University Hospital, München, Germany. [5] Institute of Cardiovascular Physiology and Pathophysiology, Biomedical Center, Planegg-Martinsried, Munich, Germany. [6] Department of Otorhinolaryngology, Ludwig-Maximilians-Universität Munich, Munich, Germany. [7] Department of Cell Biology, Johns Hopkins University School of Medicine, 855 North Wolfe Street, Baltimore, MD 21205, USA. [8] Institute of Science and Technology (IST) Austria, 3400 Klosterneuburg, Austria. [9] Institute of Experimental Biomedicine I, University Hospital and Rudolf Virchow Center, Würzburg, Germany. ✉email: steffen.massberg@med.uni-muenchen.de; florian.gaertner@ist.ac.at

Circulating platelets are anucleate cell fragments released from large precursor cells termed megakaryocytes[1]. Platelets preserve vascular integrity through hemostatic clot formation, but they are also central coordinators of vascular inflammation and host defense by directly fighting pathogenic invaders and by orchestrating leukocyte trafficking[2–6]. Inflammation triggers a canonical platelet effector program distinct from classical hemostasis and thrombosis. While clotting requires platelets to engage in multicellular aggregation and retraction of the thrombus, they mainly act as individual cells when encountering inflammation[7–10]. Under inflammatory conditions, single platelets occupy strategic positions where they serve as landing pads for leukocytes and aid in intravascular crawling and extravasation. At the same time, platelets help to limit inflammatory collateral damage. In particular, they play a central role in plugging holes in the vasculature caused by transmigrating leukocytes[2,8,11]. This ability of single platelets to seal inflammatory microlesions smaller than the diameter of a cell is essential to prevent severe bleedings in inflammation, infection, and cancer[12,13]. For efficient sealing of microlesions, individual platelets need to detect defects that are few μm$^2$ in size in an endothelial layer that covers a surface of >1000 m$^2$ in humans. Effective recruitment of inflammation-responsive platelets to microlesions therefore requires a high degree of precision, suggesting a spatiotemporally tightly coordinated rather than stochastic process.

Directional migration is a fundamental cell-biological process enabling cells to locomote along guidance cues to autonomously change position during embryonic development, cancer dissemination, and inflammation[14]. Directional locomotion does not rely on the presence of the cell nucleus, as recently demonstrated by enucleation of fibroblasts and endothelial cells in vitro[15]. Platelets are a unique physiological model of anucleate cell migration in vivo and their ability to migrate was shown to boost inflammation in sepsis[16]. However, whether platelet migration is directionally instructed by specific guidance cues from the inflammatory microenvironment and may target platelets to vascular microlesions remains unknown.

Cells employ actin-rich protrusions to explore and interact with the microenvironment and to navigate within tissue[17–20]. When platelets bind to adhesive substrate, their cytoskeleton undergoes rapid rearrangement and forms a broad circular branched actin network, the lamellipodium, resulting in a prototypic fried egg-like shape of spread platelets in vitro[21]. However, since the abrogation of lamellipodia formation does not affect hemostatic or thrombotic platelet functions[22,23,24] and visual evidence of lamellipodia in vivo has so far not been provided, the physiological role of platelet lamellipodia formation remains enigmatic.

Adhesive cells employ Arp2/3-dependent lamellipodia to establish integrin-mediated adhesions with the substrate and to migrate along adhesive gradients[25–27]. This paradigm of integrin-mediated directed cell migration along adhesive gradients is referred to as haptotaxis[28]. While various studies confirmed the crucial role of Arp2/3 in haptotaxis in vitro[25,29], it still remains unclear if branched actin networks are required for haptotaxis in vivo[30].

Here, we establish platelets as a reductionist, nucleus-free, model system to study the role of branched actin networks in integrin-mediated directional motility. We show that platelets require Arp2/3-dependent branched actin networks to directionally migrate along adhesive gradients. Platelet haptotaxis plays a critical role in vascular surveillance in vivo preventing microbleeds and bacterial invasion during inflammation and infection, while being dispensable for classical hemostasis. Our findings therefore not only identify Arp2/3-dependent haptotaxis

as a crucial effector and intrinsic surveillance function of immune-responsive platelets in vivo, but also functionally establish the role of branched actin networks in integrin-mediated directional motility within living organisms.

## Results

### Vascular fibrin(ogen) deposits instruct platelet shape and motility in inflammation

To study morphodynamics of single platelets in inflammation, we used multicolor 4D intravital spinning-disk microscopy of the cremaster microvasculature following exposure to lipopolysaccharide (LPS) (Supplementary Fig. 1a and Supplementary Methods). Inflammation triggered the accumulation of individual platelets in the microcirculation (Fig. 1a). Platelets either transiently or firmly adhered to the vessel wall and a fraction of adherent platelets subsequently migrated autonomously, a motility pattern distinct from platelets piggybacked by migrating leukocytes (Fig. 1b, Supplementary Fig. 1b–d, and Supplementary Movie 1).

Interestingly, adherent and migrating platelets colocalized with points of neutrophil extravasation (Supplementary Fig. 1e, f), the predilection sites of inflammatory bleeding, which is suppressed by platelets[7].

To visualize morphological adaptations of single platelets encountering the inflammatory microenvironment in situ, we established an optimized fixation and super-resolution microscopy protocol (see Supplementary Methods). We recently identified the acute-phase protein fibrin(ogen) as a critical adhesive ligand involved in platelet migration in vitro[16]. Indeed, when we examined the inflamed cremaster microvasculature, we detected a thin layer of fibrin(ogen), providing a two-dimensional platform for adhesion of single platelets in vivo (Fig. 1c). In situ morphological analysis of adherent platelets revealed the presence of sheet-like lamellipodial protrusions resembling the phenotype of platelet spreading and migration on a fibrin(ogen) layer in vitro (Fig. 1d and Supplementary Fig. 1g). In contrast, we did not detect lamellipodia formation in platelets recruited to the core of fibrin(ogen)-rich thrombi that form after vessel injury (Fig. 1e). In thrombi, platelets constituted a round shape with small spiky protrusions reminiscent of filopodia (Fig. 1e and Supplementary Fig. 1h)[24,31]. Unlike the two-dimensional (2D) fibrin(ogen) patches deposited at the inflamed vessel wall, fibrin(ogen) in the core of thrombotic lesions formed a three-dimensional (3D) meshwork fully imbedding platelets (Fig. 1f, Supplementary Fig. 1i, and Supplementary Movie 1). Contractile platelets exerted isotropic pulling forces on formed thrombi, collectively leading to thrombus retraction (Supplementary Fig. 1j)[32]. This collective process was distinct from the exploratory behavior of platelets migrating individually in the inflamed vasculature (Fig. 1g, Supplementary Fig. 1k–m, and Supplementary Movie 1) or at the periphery of vascular injuries[16]. Together, we identified distinct patterns of fibrin(ogen) deposition during inflammation compared to thrombosis, which triggered a characteristic morphology and motility of individual platelets (Fig. 1g, h).

### Platelets adapt to the mechanical composition of fibrin(ogen) deposits

Based on these observations, we hypothesized that platelets scan for fibrin(ogen) deposits in vivo and in turn adapt their effector functions according to the physical presentation of the ligand at the vascular lesion. To verify this, we coated coverslips with fluorescently labeled fibrinogen in the absence or presence of thrombin to generate thin fibrin(ogen) matrices with or without 3D complexity (Fig. 2a, Supplementary Fig. 2a, and Supplementary Methods).

Cross-linked fibrillar fibrin networks fostered a non-polarized, round morphology and platelets remained stationary, while

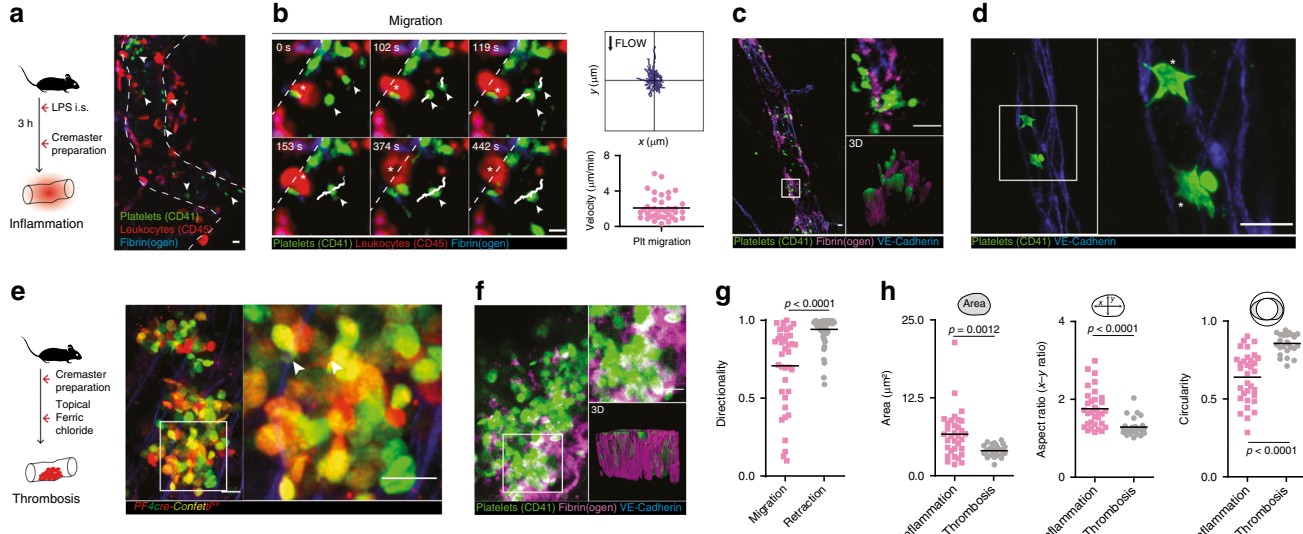

**Fig. 1 Platelets recruited to inflamed blood vessels are migratory and form lamellipodia. a**, **b** Platelet recruitment and migration in inflamed microvessels. **a** Representative micrograph and **b** platelet migratory behavior, exemplary observed patterns, tracks (white) and cell velocities ($n = 39$ cells in five mice). White arrows: platelets. **c**, **f** Platelet phenotype and microenvironment in LPS-induced inflammation and ferric chloride-induced thrombosis in the cremaster microvasculature. **c** Fibrin(ogen) deposition and **d** associated platelet phenotype in inflammatory settings. Asterisks: lamellipodia. **e** Platelet phenotype and **f** fibrin(ogen) deposition in thrombosis. White arrows: filopodia. **c**, **f** Right lower insets: rendered 3D reconstructions. **g** Directionality of in vivo migrating ($n = 39$, see **b** compared to retracting cells (mesenteric thrombus formation, $n = 59$ cells in three mice), Mann–Whitney. **h** Analysis of in vivo platelet shape in the cremaster microvasculature, $n = 26$ (thrombosis), $n = 33$ (inflammation) in three mice, Mann–Whitney. Scale bars $= 5$ µm. All statistical tests are two-sided. Source data are provided as a Source Data file.

monomeric fibrin and fibrinogen surfaces triggered platelet polarization and migration (Fig. 2b and Supplementary Fig. 2b–e).

Time-lapse microscopy revealed that platelets recruited to non-cross-linked fibrin(ogen) monolayers removed ligands from the substrate, while the associated lamellipodial edges collapsed. Platelets then polarized towards adjacent areas still coated with fibrin(ogen), where lamellipodia remained stably attached to the substrate (Supplementary Fig. 2f). This mechano-probing translated into a polarized subcellular organization of the actomyosin cytoskeleton and reinforced a half-moon-like platelet shape, leading to migration resembling the morphodynamics of platelets recruited to inflamed blood vessels (Fig. 2c, d). In contrast, platelets entangled in cross-linked fibrin matrices radially aligned the fibrin meshwork, thereby triggering isotropic contraction of the matrix (Supplementary Fig. 2d–i and Supplementary Movie 2)[33,34]. This behavior is consistent with clot retraction observed during thrombosis in vivo (Supplementary Fig. 1k). Importantly, cleavage of 3D fibrillar fibrin mesh works by the serine–protease plasmin rescued polarization and migration, highlighting the ability of platelets to instantaneously adapt their motility program to the physical composition of the microenvironment (Fig. 2e, f and Supplementary Movie 2). To confirm that mechanical rather than biochemical properties of adhesive matrices are sufficient to instruct distinct platelet motility programs, we coated coverslips with integrin ligands of tunable mechanical stability (see Supplementary Methods and Fig. 2g). When pulling forces of spreading platelets were overcoming the ligand's rupture threshold platelets polarized and migrated, consistent with platelets migrating on fibrin(ogen) monolayers (Fig. 2g and Supplementary Movie 2). Hence, platelets constantly probe the mechanical properties of their microenvironment; when contractile pulling forces overcome matrix stability, platelets remove the weakened ligands, polarize, and migrate towards regions of higher matrix densities. These observations raise the fundamental question of how platelets sense fibrin(ogen) density.

**Arp2/3-dependent haptotaxis guides platelets along fibrinogen gradients.** Platelet migration on fibrin(ogen) is preceded by the formation of radially extending, spike-like filopodia, which are followed by flat sheet-like lamellipodia (Fig. 3a, b and Supplementary Movie 3). The dominant lamellipodium determines the leading edge of the now polarized platelet and is a strong predictor of the direction of migration (Fig. 3c). We therefore hypothesized a functional role of platelet lamellipodia in reading and interpreting local fibrin(ogen) densities, as previously reported for other cell types[25,26]. To test this hypothesis, we generated coverslips coated with steep gradients of high to low fibrinogen concentrations (Fig. 3d, see Methods section) and analyzed decision-making of platelets in contact with both regions. Platelets encountering the fibrinogen density interface persistently migrated from regions of low into regions of high fibrinogen densities, while changing direction upon sensing lower substrate densities (Fig. 3d). As a consequence, virtually all platelets repositioned to areas of higher substrate density (Fig. 3e).

Lamellipodia formation requires the polymerization of branched actin networks initiated by the actin-nucleating Arp2/3 complex[22,25,26]. In line with this, migrating platelets exhibited Arp2/3 enrichment at the leading edge (Fig. 3f). Following the addition of CK666, a specific Arp2/3 inhibitor[35], platelets were unable to undergo half-moon-like polarization, while spike-like filopodia remained attached to the substrate (Fig. 3g). These morphological changes were paralleled by abrogation of actin polymerization at the leading edge and arrest of migrating platelets without affecting platelet contractility and fibrin reorganization (Supplementary Fig. 3a, b and Supplementary Movie 3 and 4). Loss of lamellipodia was accompanied by a severe reduction of fibrin(ogen)—integrin engagements at the platelet periphery (Supplementary Fig. 3c) resulting in increased retrograde actin flow[36] (Supplementary Fig. 3a and Supplementary Movie 3). This indicates a role for lamellipodia in scanning for adhesive ligands at the leading edge of migrating platelets. In

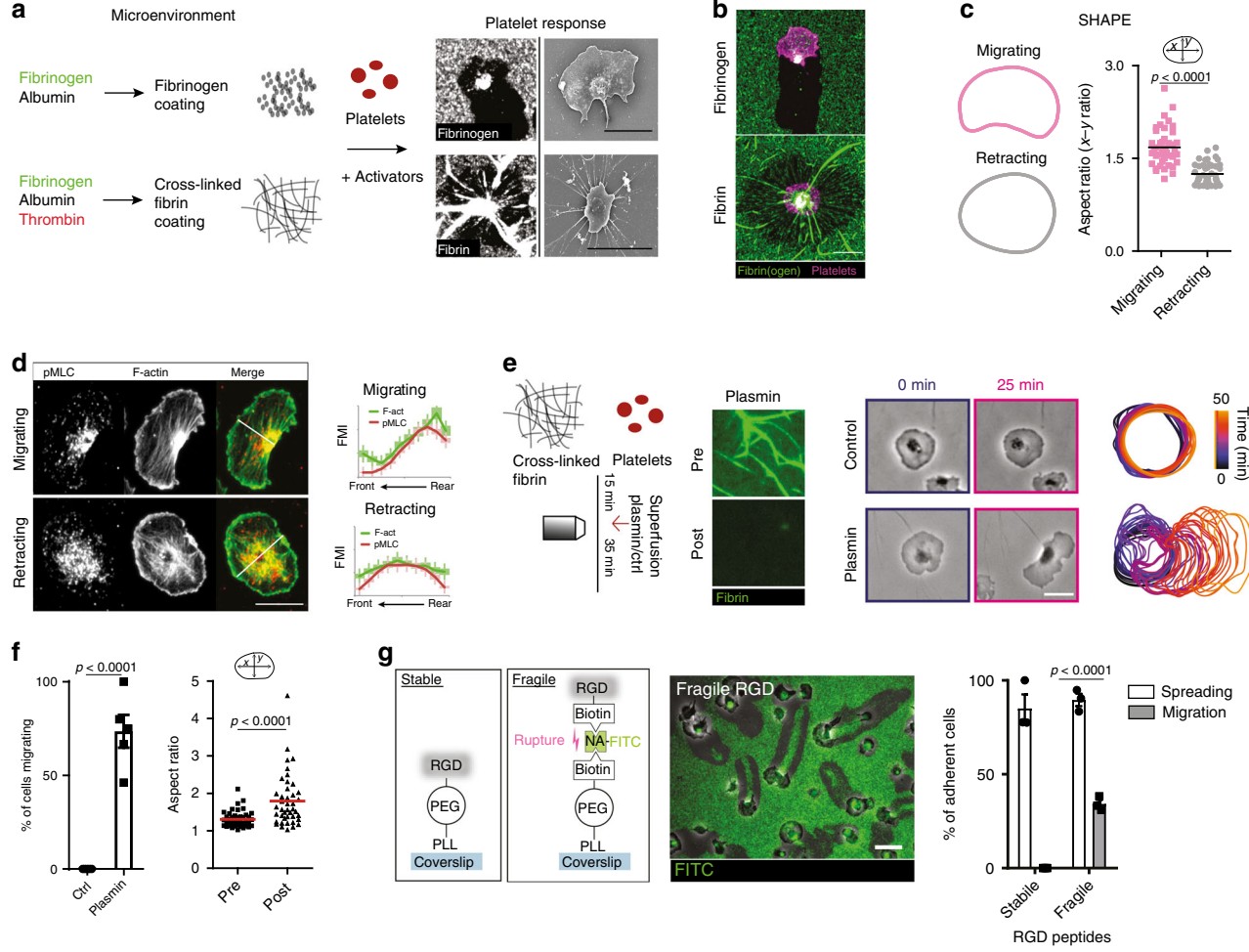

**Fig. 2 Mechanical properties of fibrin(ogen) are sufficient to instruct platelet shape and motility. a–d** In vitro reconstitution of in vivo microenvironments. **a** Substrate clearance and morphology of platelets on monomeric and cross-linked substrate. **b** Representative micrograph of migrating and retracting cell on fibrinogen and cross-linked fibrin substrate and **c** mean shape outlines and comparison of the aspect ratio of cells on fibrinogen leading to migration ($n = 43$ cells) and cells on cross-linked fibrin leading to retraction ($n = 47$ cells, Mann–Whitney). **d** p-MLC and F-actin distribution in migrating and retracting human platelets. (Left) Exemplary cells and (right) FMI profiles along short axes in $n = 50$ (migrating) and $n = 47$ (retracting) cells from three experiments. **e, f** Plasmin treatment of cross-linked substrate after human platelet addition. **e** (Left) Experimental setup and cleavage of fibers after plasmin treatment indicated by loss of fluorescence. (Center) Representative micrographs. (Right) Color-coded (time) representative cell outlines. **f** Migrating cell percentage after plasmin/control treatment ($n = 5$ experiments, Mann–Whitney), and aspect ratio prior and after treatment ($n = 46$ cells from four experiments, Wilcoxon's signed-rank test). **g** Integrin ligands of tunable mechanical stability. Left: stable: RGD (Arg-Gly-Asp)-peptides are covalently bound (stable) to a PLL (poly-ʟ-lysine)-PEG (polyethylene glycol) backbone immobilized on a glass coverslip, fragile: RGD-biotin is bound to a PLL-PEG-biotin backbone via a neutravidin-FITC (NA) bridge, middle: exemplary micrograph showing that platelet pulling forces rupture NA–biotin bonds as indicated by the loss of fluorescence (FITC) and right: percentage of spreading and migrating platelets, $n = 3$ independent experiments, $t$ test. Scale bars = 5 μm. Error bars = s.e.m. All statistical tests are two-sided. Source data are provided as a Source Data file.

addition, CK666-treated platelets adhering at the interface of high and low fibrinogen densities did not polarize nor migrate into regions of high fibrinogen, but rather symmetrically extended their filopodia into both regions (Fig. 3h, i). Hence, spreading platelets use their lamellipodia to locally sense substrate adhesiveness, eventually triggering polarization and migration in direction to the higher adhesive area, a process referred to as haptotaxis[37] (Supplementary Fig. 3d).

**Haptotactic platelets safeguard inflamed blood vessels.** To study the (patho-)physiological relevance of platelet haptotaxis in vivo, we genetically deleted the Arp2/3 complex in platelets by generating *PF4-Cre−/+; Arpc2fl/fl* mice (referred to as *Arpc2−/−*). Similar to CK666 treatment of human platelets, which preserved

normal hemostatic platelet functions, including adhesion (Supplementary Fig. 3e) and clot retraction (Supplementary Fig. 3f, g), Arpc2-deficient mouse platelets showed unaltered aggregation (Supplementary Fig. 3h), only minor changes in the activation (Supplementary Fig. 3i) and normal retraction, whereas platelet migration and spreading was abrogated (Supplementary Fig. 3j and Supplementary Movie 4). In line with the literature[22], *Arpc2−/−* mice presented normal tail bleeding times and formed stable thrombi in a model of acute arterial thrombosis similar to *Arpc2+/+* controls with platelet counts adjusted to the level of *Arpc2−/−* mice (Supplementary Fig. 4a–g, Supplementary Movie 5, and Supplementary Methods).

When we examined the inflamed cremaster microvasculature, wild-type (WT) platelets adhered to the endothelium and subsequently formed lamellipodia (see Fig. 1d), suggesting a role

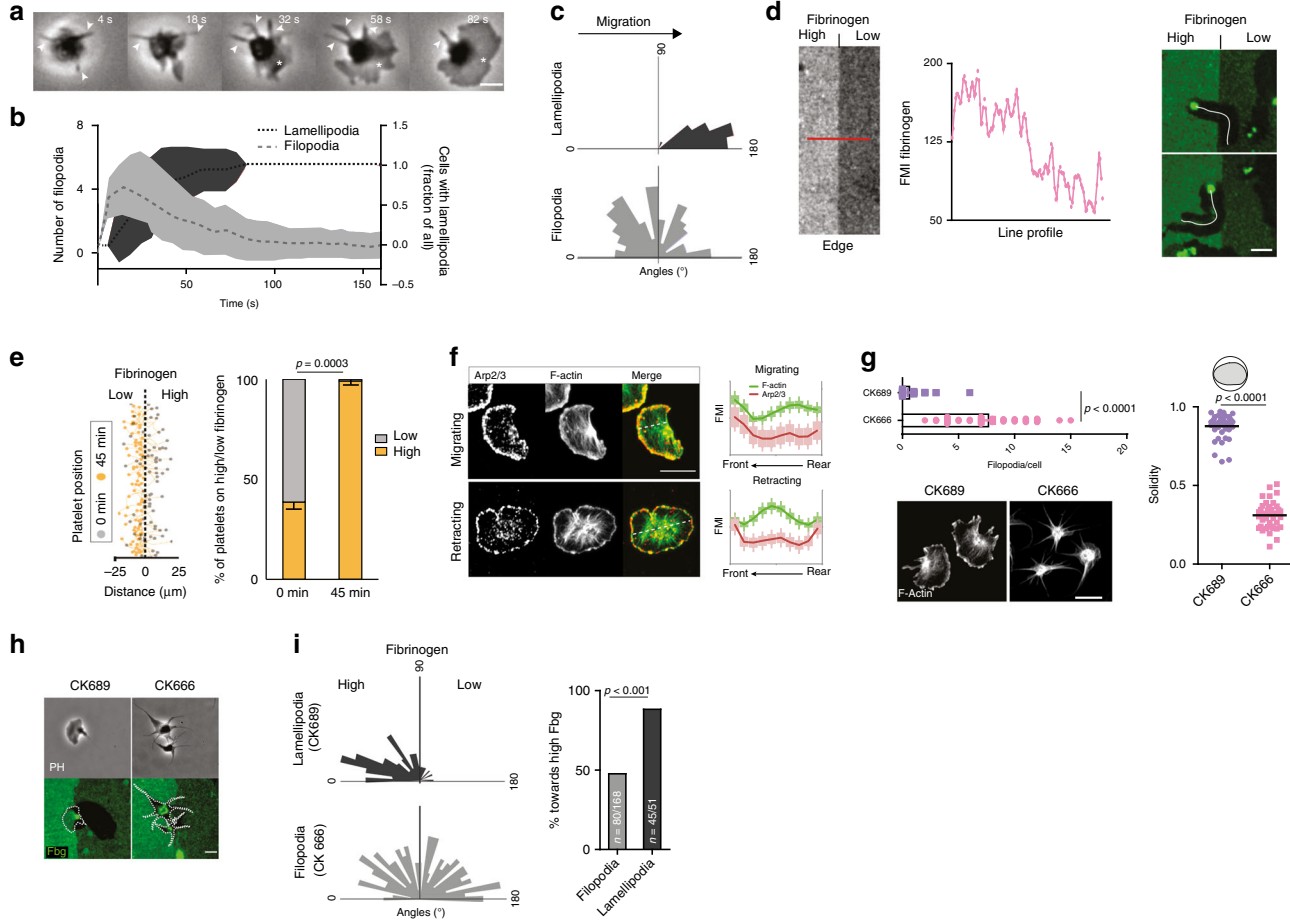

**Fig. 3 Arp2/3-dependent lamellipodia formation regulates platelet haptotaxis. a–c** Early shape changes of human platelets on fibrinogen surfaces.
**a** Micrograph of spreading cell. Arrows: filopodia; asterisk: lamellipodium. **b** Evolution of lamellipodia/filopodia over time, and **c** direction of filopodia and
lamellipodia in relation to the subsequent direction of migration. **b**, **c** $n = 32$ cells from three experiments. Lines: mean; error bands: s.d. **d**, **e** Platelet
decision-making upon encountering substrate gradients. **d** (left) Exemplary line profile of fibrinogen gradients encountered by cells and (right)
representative micrographs of the substrate (green: fibrinogen-AF488) and platelet tracks (white), 30 min after cell seeding. Scale bar = 5 μm. **e** Tracks of
cells ($n = 256$ cells) in contact with interface, and (right) platelet positioning before (0 min) and after migration of cells getting in contact with the interface
(45 min), $n = 4$ experiments, paired $t$ test. Scale bar = 10 μm. **f** Arp2/3 and F-actin distribution in migrating and retracting cells, and FMI profiles along
short axes in $n = 50$ (migrating) and $n = 53$ (retracting) cells pooled from three experiments. Scale bar = 5 μm. **g** Effect of Arp2/3 inhibition with CK666
compared to control CK689 on the number of filopodia per cell (Mann–Whitney), solidity ($t$ test), lamellipodia formation, and F-actin distribution (compare
Fig. 1d). $n = 41$ cells from three experiments. Scale bar = 5 μm. **h**, **i** Decision-making upon encountering substrate gradients, control (CK689) and Arp2/3
inhibited (CK666) platelets (**h**) representative micrograph. Scale bar = 5 μm and **i** direction of lamellipodia and filopodia in relation to the fibrinogen
gradient, $n = 168$ filopodia and $n = 51$ lamellipodia, $\chi^2$ test. Error bars = s.e.m. Scale bars = 5 μm. All statistical tests are two-sided. Source data are
provided as a Source Data file.

of spreading and haptotaxis in the context of inflammation. In
striking contrast, $Arpc2{-}/{-}$ platelets failed to form lamellipodia,
but rather developed prominent filopodia, leading to spiky cell
contours in vivo (Fig. 4a, b), consistent with our in vitro adhesion
assays (Supplementary Fig. 3j). Importantly, these morphological
alterations were accompanied by a significant reduction in
platelet migration, resulting in an increased fraction of non-
motile adherent cells, while rolling and leukocyte interaction/
motility remained unaffected (Fig. 4c and Supplementary
Movie 5).

Platelets prevent inflammatory bleeding from vascular lesions
caused by extravasating leukocytes[7,8] (Supplementary Fig. 5a–e).
To achieve this, single platelet form sealing plugs precisely
positioned to cover endothelial microbreaches[3,13]. We therefore
hypothesized that directional spreading and migration towards
higher fibrin(ogen) densities may play a role in this process.
Indeed, fibrin(ogen) was enriched at endothelial cell junctions
(Fig. 4d), the major exit routes of transmigrating leukocytes[38]. In

$Arpc2{+}/{+}$ mice, we found a large fraction of adherent platelets
positioned in close contact with endothelial cell junctions (Fig. 4e).
In addition, platelets colocalized with areas of low pericyte
coverage (Fig. 4f) that constitute weak spots of the vessel wall
often penetrated by transmigrating leukocytes[39].

$Arpc2$-deficient platelets were less efficiently recruited to spots
of leukocyte transmigration (Fig. 4e, f), suggesting a role of fibrin
(ogen) haptotaxis in preserving vascular integrity. Consistent with
previous reports, we did not observe macroscopic bleeds[22,40];
however, $Arpc2{-}/{-}$ mice showed increased inflammatory
microbleeds (Fig. 4g), despite comparable platelet numbers and
leukocyte recruitment patterns (Supplementary Fig. 5f–i). Like-
wise, interference with platelet-fibrin(ogen) engagement by
treating mice with integrin $\alpha_{IIb}$ inhibitor tirofiban resulted in
comparable microbleeds (Supplementary Fig. 5j), while macro-
scopic bleeds were absent, as previously reported for the skin[40,41].
We also observed increased microbleeds in $Arpc2{-}/{-}$ in sterile
inflammation elicited by CCL-2 and causing extensive

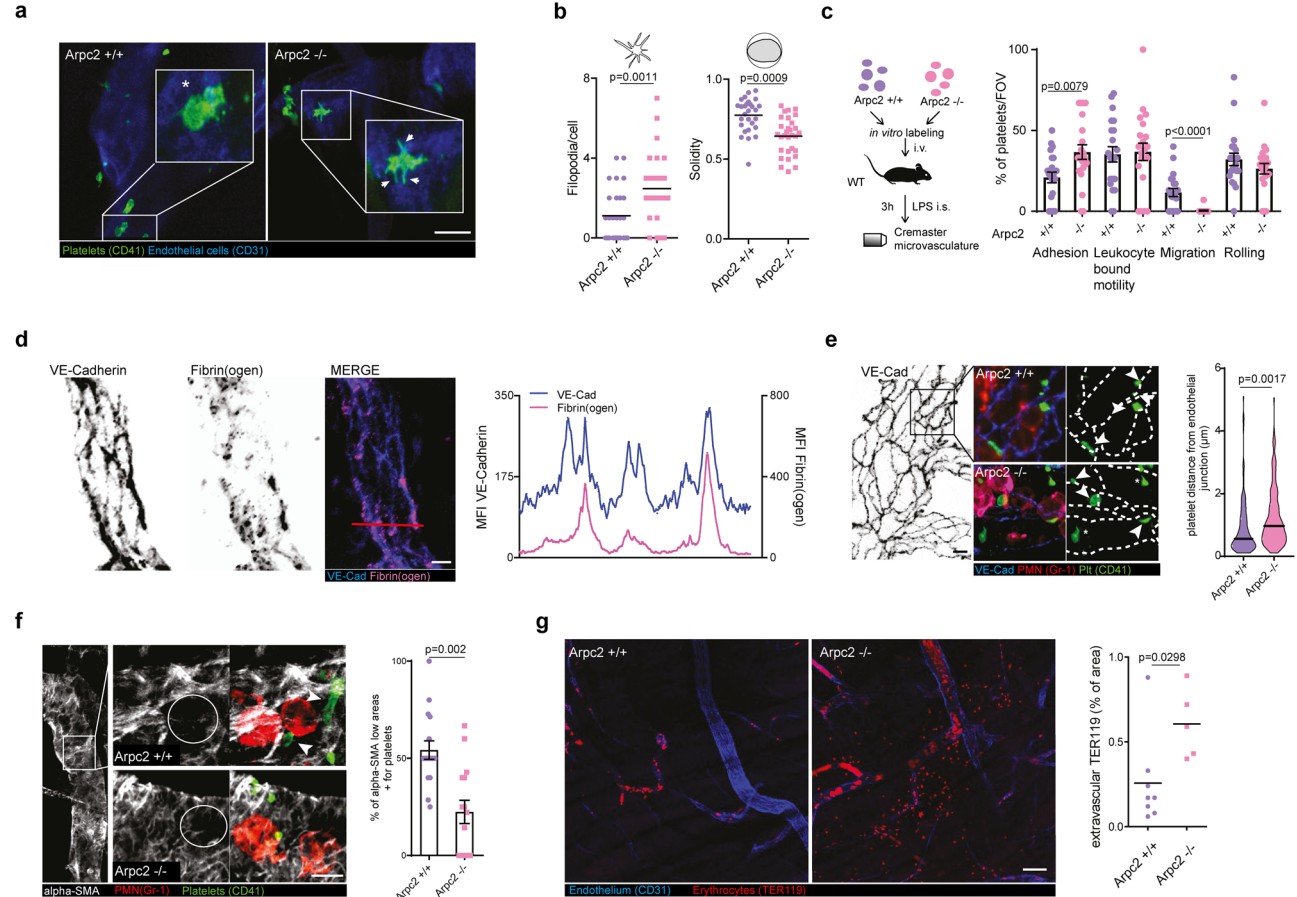

**Fig. 4 Platelet spreading and haptotactic migration preserve vascular integrity in inflammation. a, b** Analysis of Arpc2+/+ and Arpc2−/− platelet morphology in the inflamed microvasculature. **a** Cremaster whole mount of adherent platelets. Asterisk: lamellipodium; arrows: filopodia. **b** Filopodia per cell and solidity, Arpc2+/+: n = 28 cells, Arpc2−/−: n = 30 cells in three mice per group, t tests. Scale bar = 5 μm. **c** Motility patterns of transfused Arpc2+/+ and Arpc2−/− platelets. Quantification of patterns (n = 21 vessels from three mice), Mann–Whitney. **d** Distribution and MFI profile of vessel-deposited fibrinogen in relation to VE-Cadherin+ endothelial junctions. Scale bar = 5 μm. **e** Localization of Arpc2+/+ and Arpc2−/− platelets in relation to VE-Cadherin+ endothelial junctions. (Left) Exemplary whole mounts, arrows indicate platelets adherent to junctions, asterisks highlight platelets distant from junctions and (right) platelet distance from junctions, violin plot, Arpc2+/+: n = 135, Arpc2−/−: n = 144 cells from three mice per group, t test. Scale bar = 10 μm. **f** Platelet recruitment to α-SMA low regions in Arpc2+/+ and Arpc2−/− mice. (Left) Whole mount and (right) percentage of α-SMA low areas positive for platelets, Arpc2+/+: n = 16, Arpc2−/−: n = 16 vessels from three mice per group, t test. Scale bar = 5 μm. White arrows: platelets. **g** Bleeding assessment in the inflamed cremaster microvasculature of Arpc2+/+ and Arpc2−/− mice, representative whole mounts stained for erythrocytes (TER119) and vasculature (CD31) and quantification of extravascular TER119 signal, Arpc2+/+: n = 8, Arpc2−/−: n = 5 mice per group, t test. Scale bar = 50 μm. Error bars = s.e.m. All statistical tests are two-sided. Source data are provided as a Source Data file.

extravasation of neutrophils in the cremaster muscle, highlighting a general role of platelet migration in the prevention of microbleed independent of stimulus (Supplementary Fig. 5k).

Similar to the inflamed muscle tissue, we also observed lamellipodia-forming platelets in contact with fibrin(ogen) deposits in areas of vascular damage in a clinically relevant model of LPS-induced acute respiratory distress syndrome (ARDS) (Fig. 5a, b and Supplementary Methods). Arp2/3-dependent positioning of platelets played a crucial role in preventing inflammatory bleeding in the inflamed lung (Fig. 5c–f and Supplementary Fig. 6a–k). While control mice showed only mild intra-alveolar bleedings, Arpc2−/− mice developed severe pulmonary hemorrhage (Fig. 5c–f and Supplementary Fig. 6a–k). Consequently, Arpc2−/− mice suffered from impaired oxygen saturation and higher mortality (Fig. 5g, h and Supplementary Fig. 6l). Finally, and in contrast to transient pharmacological inhibition[41,42], genetic deletion of integrin αIIb, the major fibrinogen receptor, phenocopied the bleeding tendency of Arpc2−/− mice in the lung (Supplementary Fig. 6m), as also

reported for the inflamed brain microvasculature[43]. This further indicates a crucial role of fibrin(ogen)-dependent platelet spreading and migration in safeguarding the inflamed microcirculation. Taken together, Arp2/3-dependent spreading and haptotaxis constitute crucial platelet functions required for safeguarding the inflamed microvasculature in the muscle and lung and preventing inflammatory microbleeds.

**WASp and WAVE play a redundant role in platelet haptotaxis.** The Arp2/3 complex of platelets is activated by nucleation-promoting factors (NPFs) of the Wiskott–Aldrich syndrome protein (WASp) and WASp-family verprolin-homologous protein (WAVE) families[17,27]. To test the contribution of these NPFs in platelet migration and haptotaxis, we analyzed platelets deficient in WASp or Cyfip1, the Rac1-binding subunit of the WAVE complex required for its activation in platelets[24]. WASp−/− platelets formed lamellipodia similar to control platelets and showed unimpaired migration and fibrinogen removal, indicating

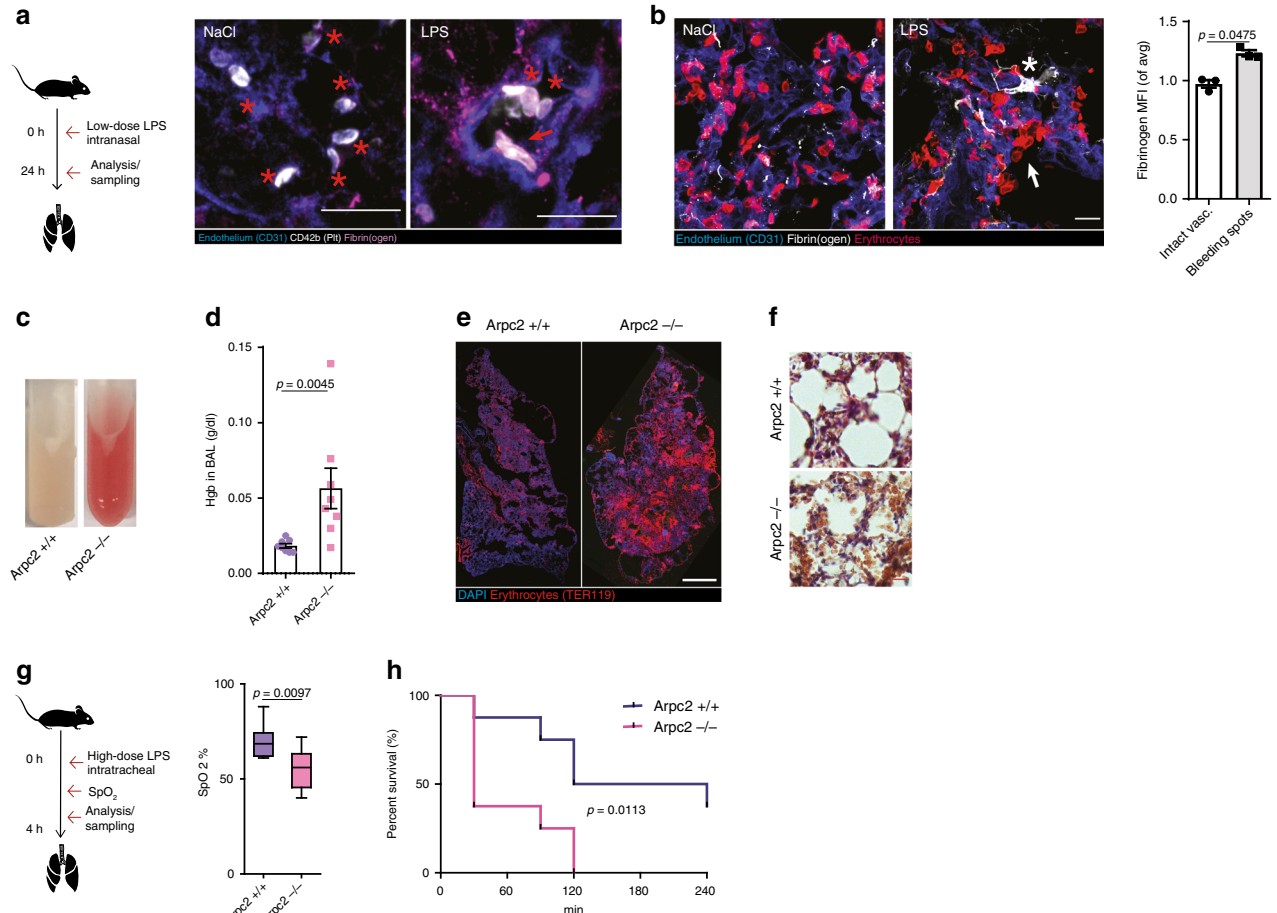

**Fig. 5 Vascular surveillance by platelets prevents pulmonary bleeding. a**, **b** LPS-induced (sub-)acute lung injury model. **a** Representative micrograph of fibrinogen deposition (asterisks) and platelet morphology in the inflamed (LPS) and control lungs (NaCl) of WT animals. Asterisk: lamellipodium. Scale bar = 5 μm. **b** Exemplary micrograph and quantification of fibrin(ogen) deposition at sites of vascular damage (arrows) compared to control mice indicated by single extravasated erythrocytes and intact vasculature, $n = 3$ WT mice, paired $t$ test. Scale bar = 5 μm. **c**, **f** LPS-induced lung injury in busulfan-treated *Arpc2+/+* and *Arpc2−/−* mice. **c** Appearance, **d** hemoglobin content in bronchoalveolar lavage (BAL), and **e** lung histology 8 h after LPS challenge. Scale bar = 10 mm. **f** Lung HE stain. Scale bar = 30 μm. **d** Mann–Whitney. **g**, **h** Hyperacute lung injury model. **g** SpO₂ 90 min after high-dose tracheal LPS application. Center: mean; whiskers: min–max; box: 25–75th percentiles. **h** Kaplan–Meier survival curve, $n = 8$ mice per group. **g** $t$ Test and **h** log rank. Error bars = s.e.m. All statistical tests are two-sided. Source data are provided as a Source data file.

a redundant role of WASp in lamellipodia formation and platelet migration (Fig. 6a). Previous reports have shown that WAVE rather than WASp is required for lamellipodia formation in platelets[24,44]. Indeed, spreading area of platelets isolated from conditional *Pf4cre+/− Cyfip1fl/fl* (*Cyfip1−/−*) mice was largely reduced compared to controls (Fig. 6b) and adherent platelets formed prominent filopodia (Fig. 6b), confirming previous observations[24]. Time-lapse video microscopy, however, revealed premature lamellipodia-like extensions that protrude and retract between adjacent filopodia, but do not extend to filopodial tips (Supplementary Movie 6); and mean projected platelet area was significantly larger compared to *Arpc2−/−* platelets (Fig. 6c–e). *Cyfip1−/−* platelets showed a slight, but not significant reduction of migration and unimpaired removal of fibrinogen, indicating that premature lamellipodia-like extensions were sufficient to promote platelet migration (Fig. 6f, g). Incubation of *Cyfip1−/−* platelets with the Arp2/3 inhibitor CK666 abolished migration, indicating sufficient residual Arp2/3 activation at the leading edge of migrating *Cyfip1−/−* platelets (Fig. 6h). Hence, *Cyfip1−/−* platelets were able to haptotax and preferentially migrated to higher fibrinogen densities (Fig. 6i). During ARDS, *Cyfip1−/−* platelets prevented inflammatory bleedings of the lung (Fig. 6j),

despite the reduction of platelet counts to the level of *PF4-Cre; Arpc2fl/fl* mice (Fig. 6k), indicating functional haptotaxis in vivo. Consequently, and similar to previous reports of other cell types[45], our data suggest a partial redundancy of NPFs in the activation of the Arp2/3 complex, which is sufficient to promote migration when one or the other is deleted.

**Vascular surveillance by lamellipodia-forming platelets prevents bacteria dissemination within the host.** We recently identified platelet migration as a platelet function facilitating platelet-pathogen interactions[16]. To gain access to the bloodstream pathogens are known to exploit vascular breaches[46]. We, therefore, hypothesized that haptotactic positioning of platelets at vascular microlesions may serve as a physiological line of defense capturing invading pathogens at primary sites of infection such as the lung[5,47,48].

*Methicillin-resistant staphylococcus aureus* (MRSA) is a major cause of hospital-acquired pneumonia[49,50] and platelets efficiently sense, bind and interact with MRSA[51]. Consequently, platelets were recruited to MRSA-containing biofilms (Fig. 7a–c). Once activated, platelets spread and migrated, thereby collecting and bundling MRSA on their surface, which led to the partial

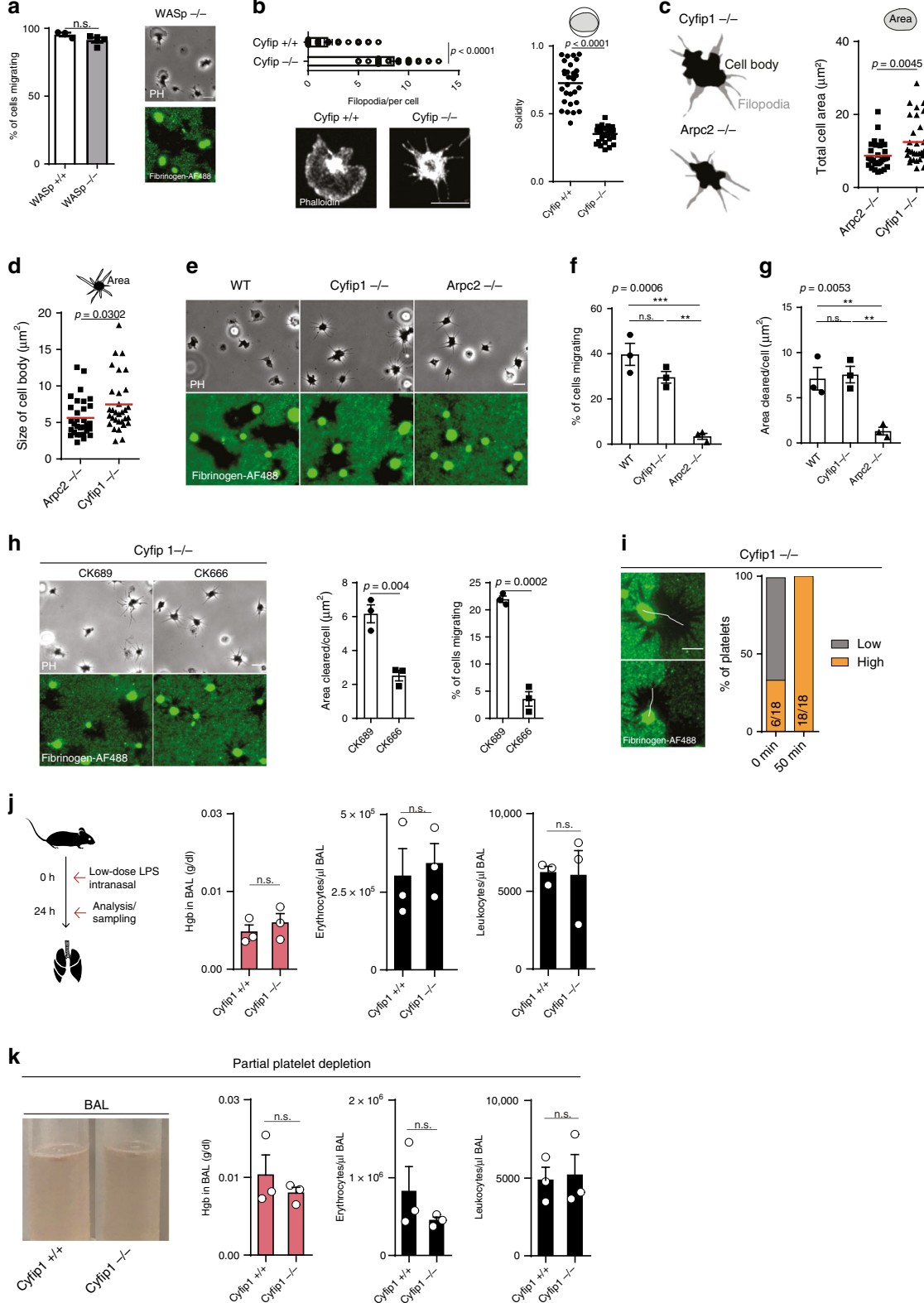

killing of bundled bacteria (Fig. 7a–c and Supplementary Fig. 7a–d and Supplementary Movie 7)[51]. Inhibition of Arp2/3 reduced collection and bundling of *MRSA* and attenuated platelet-mediated killing of biofilm-associated pathogens (Fig. 7b and Supplementary Fig. 7e).

Next, we inoculated mice with *MRSA* to assess platelet–bacteria interactions in Gram-positive pneumonia (Fig. 7d and Suppl-

ementary Methods). After 48 h of infection, lungs were over-grown with *MRSA*, largely accumulating in neutrophil-rich abscesses, with a fraction of bacteria disseminated in the lung vasculature (Supplementary Fig. 7f–g). We detected non-aggregated platelets mainly outside abscesses, where they formed lamellipodia and bundled bacteria that had escaped to non-abscessing lung tissue (Fig. 7d–f and Supplementary Fig. 7g).

**Fig. 6 WASp and WAVE play a redundant role in platelet haptotaxis. a** Morphology and migration of *WASp−/−* platelets compared to controls, *n* = 3 *WASp+/+* and *n* = 4 *WASp−/−* mice, *t* test. Scale bar = 5 μm. **b** Number of filopodia per cell (Mann–Whitney), solidity (*t* test), and representative F-actin-stained platelet of *Cyfip1−/−* (*n* = 32 cells) and *Cyfip1+/+* (*n* = 30 cells) mice, three mice per group. **c**, **d** Representative cell body/filopodia outline and **c** total cell area of *Arpc2−/−* (*n* = 31 cells) and *Cyfip1−/−* mice (*n* = 31 cells), and **d** size of cell body, three mice per group, *t* test. **e** p-MLC and F-actin distribution of *Arpc2−/−* (*n* = 39 cells) and *Cyfip1−/−* mice (*n* = 39 cells), three mice per group. Insets: F-actin distribution between filopodia. Scale bar = 5 μm. **e–g** Migration of WT, *Cyfip1−/−*, and *Arpc2−/−* platelets, **e** representative micrographs, **f** percentage of cells migrating and **g** area cleared per cell, ANOVA and Tukey's post hoc test, *n* = 3 mice per group. Scale bar = 5 μm. **h** Effect of additional Arp2/3 inhibition on *Cyfip1−/−* platelets. Representative micrographs, and analysis of percentage of cells migrating and area cleared per cell of CK689- and CK666-treated *Cyfip1−/−* platelets, *n* = 3 mice per group, paired *t* test. **i** Decision-making upon encountering substrate gradients, (left) examples of *Cyfip1−/−* platelets in contact with interface, and (right) platelet positioning before (0 min) and after migration (45 min), *n* = 18 cells from three mice. **j** LPS-induced lung injury in *Cyfip1+/+* and *Cyfip1−/−* mice, quantification of hemoglobin content, erythrocytes, and leukocytes in bronchoalveolar lavage (BAL), *n* = 3 mice, *t* test. **k** LPS-induced lung injury in busulfan-treated, partially platelet-depleted *Cyfip1+/+* and *Cyfip1−/−* mice. Quantification of hemoglobin content, erythrocytes, and leukocytes in bronchoalveolar lavage (BAL), *n* = 3 mice. Scale bars = 5 μm, *t* test. Error bars = s.e.m. All statistical tests are two-sided. Source data are provided as a Source data file.

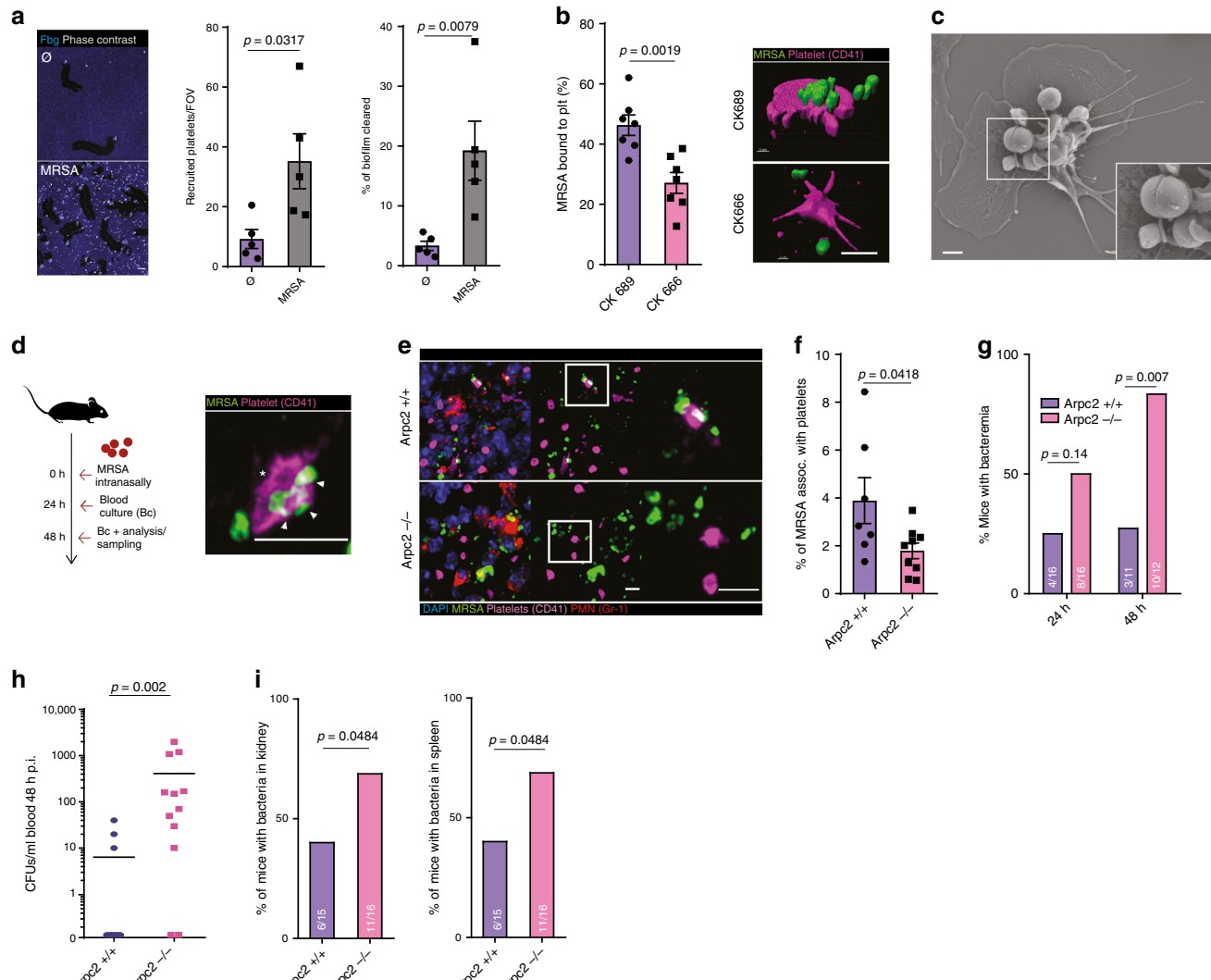

**Fig. 7 Migrating platelets protect from bacterial dissemination in MRSA pneumonia. a** Platelet recruitment and clearance in sterile and MRSA biofilms in vitro, *n* = 5 experiments, Mann–Whitney. Scale bar = 10 μm. **b** In vitro effect of Arp2/3 inhibition with CK666 compared to control CK689 in platelet-mediated MRSA biofilm clearance. (Left) Percentage of MRSA associated with platelets, *n* = 7 experiments, *t* test, and (right) 3D reconstruction of platelets interacting with MRSA. Scale bar = 5 μm. **c** Scanning electron microscopy of platelet surface-bound MRSA. Inset: bundled bacteria. Scale bar = 1 μm. **d–i** MRSA lung infection model of *Arpc2+/+* and *−/−* mice, (**d**) (left) mice were intranasally inoculated, and (right) exemplary micrograph of platelet–MRSA interaction in the lung of *Arpc2+/+* mice. Asterisks: lamellipodium; arrows: bundled bacteria. **e** Micrographs of platelet–MRSA localization and **f** percentage of MRSA associated with platelets in non-abscessing infected lung tissue (*Arpc2+/+*: *n* = 7, *Arpc2−/−*: *n* = 9 mice per group). **g** Percentage of mice with bacteremia was assessed 24 h (*n* = 16) and 48 h p.i. (*Arpc2+/+*: *n* = 11, *Arpc2−/−*: *n* = 12 mice per group). **h** MRSA CFUs recovered from blood at 48 h p.i. (**i**), percentage of mice with MRSA in the kidney and spleen (*Arpc2+/+*: *n* = 15, *Arpc2−/−*: *n* = 16 mice). **d**, **h** Mann–Whitney and **g**, **i** $\chi^2$. Scale bars = 5 μm. Error bars = s.e.m. All statistical tests are two-sided. Source data are provided as a Source data file.

Platelet–bacteria interactions in the vasculature were significantly reduced in *Arpc2*−/− mice, indicating an important role for platelet spreading and migration in this process (Fig. 7e, f). Importantly, the number of recruited platelets and neutrophils, as well as neutrophil activation and abscess formation, was not affected by the loss of *Arpc2* in platelets (Supplementary Fig. 7h–k). To determine whether impaired bacterial capture by platelets had an impact on bacterial dissemination within the organism, we quantified *MRSA* in the blood and at distant organ sites 24–48 h following a lung infection. While bacterial loads within the lung remained unaffected, *Arpc2*−/− mice were prone to developing bacteremia, resulting in an increased fraction of mice with disseminated *MRSA* in the kidney and spleen (Fig. 7g–i and Supplementary Fig. 7l). Consequently, platelets use their ability to spread and directionally migrate to capture invasive bacteria at the primary foci of infection, thereby preventing dissemination within the host.

## Discussion

Since the morphological characterization of platelet spreading in the 1970s[21,52], platelet lamellipodia formation has evolved a reductionist model to study the actin cytoskeleton and adhesion formation[53–55]. Yet, its physiological function remained largely enigmatic[22,23,24]. Our study now provides evidence of a functional role of branched actin networks of platelet in vivo. Spreading platelets use lamellipodial protrusions to scan for adhesive cues within the inflammatory microenvironment, allowing individual platelets to haptotax and optimize their position at the vascular wall.

Soluble fibrinogen is a highly abundant plasma protein (2–4 mg/ml) that becomes immobilized at thrombotic lesions, but also at sites of inflammation[56]. We show here that during inflammation vascular fibrinogen deposits flag up and tag weak points in the inflamed endothelium. Surface-anchored fibrin(ogen) is recognized and bound by platelet integrin $\alpha_{IIb}\beta_3$[57]. Mechanosensing platelets in turn use integrins to transmit forces from the actomyosin cytoskeleton to the substrate[58] and subsequently adapt their effector functions to the mechanical properties of the microenvironment[33,34]. When the cytoskeleton–integrin–substrate linkage overcomes a critical tension threshold integrins become activated (outside–in signaling) and platelets firmly adhere[59,60]. In contrast, a fragile substrate rupturing below this critical tension threshold impedes integrin activation[59]. Thus, mechanical substrate properties play a critical role in initiating platelet activation and adhesion. Here, we now report a second mechanical checkpoint allowing activated, adhesive platelets to switch from isotropic contraction to anisotropic migration. When pulling forces generated by adhesive platelets overcome substrate stability, platelets start to pile-up the weakened ligands and scan the microenvironment to establish new adhesive bonds. Platelets then preferentially migrate towards areas of higher integrin-ligand density, a process referred to as haptotaxis[37].

Platelet haptotaxis is not restricted to preformed density gradients of integrin ligands. Adhesive platelets generate high traction forces that instantaneously deplete weak integrin ligands from the underlying substrate and form steep, self-generated haptotactic gradients. This ability allows platelets to locally and dynamically respond to changes of substrate mechanical properties and to migrate to areas of high ligand stability, as observed during fibrinolysis of stable fibrin networks. The concept of self-generated gradients has been described for chemotactic cells migrating in homogeneous chemokine solutions where binding of the chemotactic ligand to dummy receptors sequesters and removes the ligand and serves as a sink to generate local chemokine gradients[61]. Together, our data highlight single platelets as autonomous mechanosensors that, in addition to mechanical

remodeling of the substrate[33], use integrins to probe their adhesive microenvironment and to optimize their own positioning.

In line with previous studies in other adhesive cell types, we show that two-dimensional, sheet-like lamellipodia, but not one-dimensional, needle-like filopodia, are the mechanosensitive platelet protrusions required for the spatial integration of signals from cell–matrix adhesion sites[19,25,29,30]. Consequently, abrogation of branched actin networks and lamellipodia formation blocks platelet migration and haptotaxis. Importantly, Arp2/3-dependent actin nucleation was required for directional platelet migration in vivo, establishing a role of branched actin networks in directional integrin-dependent migration in living organisms. Arp2/3 in platelets is activated by NPFs of the WAVE and WASp families, as well as cortactin and HS1[24,44,62]. Patients with loss-of-function mutations of WASp present with immunodeficiency and bleeding complications, known as Wiskott–Aldrich syndrome[63]. Consequently, it was tempting to speculate whether impaired platelet migration and haptotaxis contribute to the clinical symptoms of WAS patients. Our data show that single deletion of neither WASp nor WAVE was sufficient to fully inhibit Arp2/3 activation at the leading edge and to block platelet migration and haptotaxis, suggesting redundancy of NPFs in this process[45]. In contrast to *WASp*−/−, WAVE deletion (*Cyfip1*−/−) abrogated large circular lamellipodia in platelets[24]. However, the short Arp2/3-dependent lamellipodial-like extensions forming at the center of *Cyfip1*−/− platelets, were still sufficient to promote platelet migration and haptotaxis, indicating a partial redundancy of NPFs in platelet motility. Further studies will be required to shed light on the mechanisms of Arp2/3 activation in migrating platelets and to investigate the mechanistic framework underlying the partial redundancy of NPFs.

Platelet haptotaxis is not required for acute thrombotic and hemostatic clot formation. During thrombosis and hemostasis, large numbers of platelets are simultaneously activated and embedded within a stable three-dimensional fibrin network that prevents platelet polarization and migration. In contrast, platelets recruited in inflammation employ a different strategy: they scan the mechanical properties of the substrate and follow two-dimensional haptotactic fibrin(ogen) gradients to optimize their position and to safeguard vascular microbreaches. Autonomous navigation of single platelets at sites of inflammation and infection is required to efficiently prevent inflammatory bleedings and bacterial invasion without the need of forming large platelet thrombi, which would run the risk of causing detrimental vessel occlusions[64,65] (Supplementary Fig. 7m).

In conclusion, we identified Arp2/3-dependent lamellipodia formation and haptotaxis as a distinct molecular signature of immune-responsive platelets in vivo. Platelet haptotaxis endows platelets with the ability to scan the vascular endothelium for inflammatory microlesions. Plugging of microlesions by migrating platelets helps prevent inflammatory microbleeds and suppresses bacterial dissemination within the host. Hence, platelet migration and lamellipodia formation serve two key functions in fighting infections: they not only collect bacteria that already gained access to the intravascular compartment[16], they also serve as gate-keepers of the inflamed vasculature preventing invasion of extravascular bacteria. Pharmacological manipulation of lamellipodia formation may therefore provide a strategy to address platelets' immunological functions without affecting physiological hemostasis. In addition, developing anti-platelet drugs that preserve lamellipodia formation may help to avoid unwanted immune-related side effects of anti-platelet therapy[66].

## Methods

**Mouse strains**. PF4-Cre[67], LifeAct-eGFP[68], Rosa26-Confetti[69], WASp−/−[70], and C57BL/6 mice (labeled WT mice) were purchased from The Jackson Laboratory and maintained and cross-bred at our animal facility. *IL4R/GPIb-tg* mice were a gift of Dr.

Ware[71]. $\alpha_{IIb}$ mice were a gift of Dr. Frampton[72]. *Arpc2fl/fl* mice were provided by Rong Li and the Wellcome Trust Sanger Institute[73,74]. *Cyfip1fl/fl* mice were recently described[24]. For a list of primers used for genotyping of all strains, please refer to Supplementary Data 2. All strains were backcrossed to and maintained on C57BL/6 background. Both male and female mice were used in this study. If not otherwise stated, animals of control and experimental group were sex-matched and age-matched (6–12 weeks) and randomly assigned to experimental groups, and *PF4-Cre*-negative control animals were used as controls. *PF4cre+/−  Arpc2fl/fl* mice exhibit mild thrombocytopenia[22]. To exclude differences in platelet counts as a possible confounder, platelet counts were adapted in *Arpc2+/+* control animals via partial low-dose busulfan depletion for in vivo experiments comparing *Arpc2−/−* with *Arpc2+/+* mice (see Supplementary Fig. 3k), if not indicated otherwise. For sepsis experiments, knockout (KO) and control animals were paired according to age, sex, and weight. All mice live in standardized conditions where temperature, humidity, and hours of light and darkness are maintained at a constant level all year round. All animal experiments were performed in compliance with all relevant ethical regulations for studies involving mice and were approved by the local legislation on protection of animals (Regierung von Oberbayern, Munich). For a detailed overview over-performed mouse experiments, please see Supplementary Data 1.

**Human blood donors.** Human blood was drawn from male and female healthy voluntary donors at the age of 25–40 years after informed consent was obtained from all subjects. Both genders were equally represented in all our analyses. Experiments involving human subjects have been approved by the ethical review board (LMU Munich) and complied with relevant regulation for experiments involving human samples.

**Bacterial strains.** MRSA strain USA300 was cultured similarly to published protocols[16]. Briefly, bacteria were cultured overnight in brain heart infusion (BHI, BactoTM Brain Heart Infusion (Porcine), BD) medium at 37 °C with 180 r.p.m. shaking. The CFU (colony-forming units) per milliliter (ml) of the overnight culture was determined for OD600 = 2 with $6.7 \times 10^8$ CFU/ml and was directly used for in vitro experiments. For infection experiments, the overnight culture was diluted in BHI to OD600 = 0.1 and further incubated at 37 °C to an OD of 2 (~4.5 h), corresponding to $6.7 \times 10^8$ CFUs/ml. Bacteria were centrifuged, and the final infection dose was diluted in 30 µl of 0.9% sodium chloride (NaCl) solution (Braun) and kept on ice until injection.

**Antibody and reagents**
*Antibodies.* For IVM (in vitro maturation), flow cytometry and histology/immunofluorescence staining the following dyes were utilized: fluorescein isothiocyanate (FITC), phycoerythrin, allophycocyanin (APC), eFluor660, and AlexaFluor 488, 594, and 647. Used antibody clones were anti-mouse Ly6G/C (clone RB6-8C5), anti-mouse CD31 (MEC13.3), anti-mouse Ly-76(TER119), anti-mouse GpIb (X488/X649, derivatized, polyclonal, Emfret), anti-mouse α-SMA (1A4), anti-mouse VE-Cadherin (VECD-1), anti-mouse hIL4-R (MAB230, R&D), anti-mouse p34-Arc/ArpC2 (Merck Millipore), anti-mouse Arp2 (polyclonal, ECM Biosciences), anti-mouse Arp3 (monoclonal, FMS338, Sigma), anti-mouse GPVI (HY101), anti-mouse CD62P(RMP-1), anti-mouse-activated CD41 (JonA), anti-human CD41(HIP8) anti-human-activated CD41(PAC-1), anti-human CD62P (AC1.2), anti-human CD42b(HIP1), anti-human Arp2/3 (13C9, Merck), anti-human phospho-myosin light chain 2 (p-MLC, Thr18/Ser19, Cell Signaling), anti-fibrinogen (sheep polyclonal, Bio-Rad), *S. aureus* antibody (polyclonal, Sigma). Used secondary antibodies included: goat anti-rabbit, goat anti-sheep, and rat anti-mouse (all Thermo Fisher). For platelet depletion, purified rat anti-mouse GP1bα antibodies and control immunoglobins (R300, Emfret) were used; for neutrophil depletion, anti-Ly6G (1A8) was used. Antibodies were purchased from BioLegend or BD Biosciences, if not stated otherwise. Antibodies were used 1:100, and secondary antibodies were used 1:200, if not indicated otherwise. Appropriate species, class, and subclass matched control antibodies were used.

*Inhibitors.* aIIbb3 inhibition: Abciximab (C7E3-Fab) from Janssen and Tirofiban from Ibigen, myosin IIa inhibition: blebbistatin(−) (Cayman Chemical) (50 µM); blebbistatin(+) (Cayman Chemical) was used as a negative control. To avoid fluorescence-induced toxicity, *para*-nitroblebbistatin (50 µM, Optopharma) was used in a subset of experiments. Arp2/3 complex inhibition: CK666 (200 µM) and negative control: CK689 (both Merck Millipore). Actin polymerization inhibition: cytochalasin D (2.5 µM) (Sigma). Other: carboxyfluorescein succinimidyl ester (CFSE), Annexin V, busulfan, plasmin, thrombin, Fluo-4 AM, LPS O111:B4, CCL-2 (Sigma) iron (III) chloride (FeCl₃), casein, recombinant human albumin (HSA), fibrinogen from human plasma, unconjugated and conjugated to AlexaFluor 488 and 594, Rhodamine 6G, adenosine diphosphate (ADP), NP40, polyethylene glycol 400 (PEG400), phalloidin-AlexaFluor 488, hexamethyldisilazane (HMDS), paraformaldehyde (PFA), GDA, and LIVE/DEAD BacLight Bacterial Viability Kit were purchased from Sigma-Aldrich. Horm collagen was from Takeda, prostaglandin I2 sodium salt from Abcam, and U46619 from Tocris.

**Mouse anesthesia.** If not stated otherwise, anesthesia was performed by isoflurane induction, followed by intraperitoneal injection of medetomidine (0.5 mg/kg body weight), midazolam (5 mg/kg body weight), and fentanyl (0.05 mg/kg body weight). Toe pinching reflexes and breathing pattern were used to determine the adequate depth of anesthesia. Core body temperature was maintained by heating pads, and narcosis was maintained by repetitive injections of 50% of the induction dose, if necessary.

**Spinning disc IVM of cremaster microvasculature.** To assess platelet motility patterns under inflammatory conditions, we either injected fluorescent fibrinogen (150 µg) and antibodies to label platelets (X649, 25 µl) and leukocytes (CD45-PE, 15 µl), or adoptively transferred cells, before intrascrotal LPS (50 µg O111:B4, in 400 µl NaCl 0.9%) application. After 4 h, the cremaster muscle was carefully prepared for intravital microscopy[75]. An upright spinning-disk confocal microscope (3i Examiner, Zeiss, ×20 objective, laser power 30%) was used for 4D image acquisition (512 × 512 px, z-slices of 2.5 µm thickness, acquisition 17–23 s/frame). Multiple areas of platelet recruitment were identified and imaged up to 1 h.

**Busulfan depletion.** To adjust platelet counts in *PF4-Cre*-negative *Arpc2fl/fl* (*Arpc2+/+*) or WT animals to *Arpc2−/−* levels (~30–50% reduction of platelet counts[22,76]) we used an established partial depletion model[77]. Briefly, busulfan (25 mg/ml in PEG400) was heated to 70 °C and diluted 1:10 in warm NaCl. Mice received 20 µg/g weight intraperitoneally −14 and −11 days before the experiment. This partial depletion method was performed for all in vivo experiments comparing *Arpc2+/+* and *Arpc2−/−* mice.

**Adoptive transfer**
*Imaging.* For adoptive transfer experiments, intracardially drawn blood was diluted 1:1 in modified Tyrodes buffer (pH 6.5) and centrifuged (25 min, 70 × g). The supernatant was incubated with either DyLight 488 or DyLight 649 anti-platelet antibody (Emfret, 1:100). To determine platelet counts, a fraction was measured on a hemocounter (Horiba). After 20 min, 2× excess Tyrodes buffer was added and the labeled platelets pelleted (1200 × g, 10 min). The pellet was resuspended in Tyrodes buffer to a final count of 10⁸ platelets/100 µl (injection doses). KO (*PF4cre-Arpc2*) platelets were injected in parallel with respective cre-negative control platelets, as well as phycoerythrin-conjugated CD45 antibody (12 µl per mouse) to visualize leukocytes. After tail vein injection, cremaster inflammation and imaging were performed as described in "Spinning disc IVM of Cremaster Microvasculature." Antibody labeling color was switched between experiments and the researcher who performed quantification of movement patterns was blinded for color assignment.

*Depletion and substitution.* To substitute recipient mice with transferred platelets, IL-4R/GPIb-tg mice were used[16,78]. In detail, thrombocytopenia was induced in IL-4R/GPIb-tg mice by i.v. injection of antibodies against hIL-4R (2.5 mg/g body weight; 12 and 3 h before transfusion). WPs from *Arpc2−/−* and +/+ were isolated as described above. Platelets from several donor mice were pooled, and 2 × 10⁸ platelets/mouse were injected.

**Cremaster whole-mount and super-resolution microscopy.** For whole-mount assessment of the cremaster microvasculature, mice were sacrificed 6 h after intrascrotal LPS challenge (50 µg, O111:B4) or CCL-2 challenge (0.3 µg), and after incision of the right atrium, the animal was perfused with 10 ml of 0.1% PFA in phosphate-buffered saline (PBS). To maintain platelet in vivo, morphological phenotype (filopodia and lamellipodia) mice were perfused with 10 ml of 0.1% PFA and 0.005% GDA in PBS for shape assessment experiments. For thrombosis assessment, cremaster was surgically exposed and treated with FeCl₃ solution (Sigma, 10%) for 20 min. Then, mice were perfused with 10 ml of 0.1% PFA and 0.005% GDA in PBS. Cremaster muscles were excised and fixated in either 4% PFA for 10 min or 2% PFA supplemented with 0.01% GDA-PFA for 20 min. Permeabilization was achieved by incubating the tissue for 2 h in 2% bovine serum albumin (BSA) with 0.1% NP40 and Triton X-100. Antibody staining was performed overnight with primary labeled antibodies. GDA-PFA fixation allowed for assessment of bleeding (extravascular erythrocytes outside of CD31+ vessels), as it preserved extravascular tissue integrity. For all other experiments, PFA fixation was used. After embedding in Mounting Medium (Sigma), whole mounts were assessed with a confocal microscope. For assessment of fibrin(ogen) deposition, AF594-conjugated fibrinogen was injected 10 min before LPS challenge (120 µg). To differentiate individual platelets and their morphology in thrombi, the multicolor platelet reporter mouse PF4cre+/−Rosa26-Confetti was used. For depletion experiments, WT mice received commercially available platelet depletion antibody or control (R300, or C301, Emfret) i.v. (2 µg/g)[79]. After 24 h, neutrophils were depleted by intraperitoneal injection of anti-Ly6G clone 1A8 or IgG isotype as control (100 µg/mouse)[80]. After 12 h, mice were subjected to the cremaster hemorrhage model as described. For analysis of the role of integrin $\alpha_{IIb}$ in the prevention of microbleeds, Tirofiban (0.5 µg/g body weight) or a similar amount of NaCl as control was injected prior and 3 h after LPS-induced inflammation, and bleeding was assessed at 6 h post infection (p.i.). Repeated injections were necessary due to the short half-life of the drug. Whole mounts were imaged with an inverted Zeiss LSM 880 in AiryScan Super Resolution (SR) Mode (10/0.3 obj. or ×20/0.8 obj. for overviews, ×40/1.3 oil immers. obj. for single-cell analysis). For in vivo morphology assessment, areas of high leukocyte recruitment were examined for

platelets undergoing shape change. To assess platelet shape, 3D z-stacks were recorded (0.5 μm stack interval) and thrombotic lesions were scanned in full to exclude spread of platelets at the periphery of the clot. After a maximum intensity projection (MIP) of these cells, cell outlines were traced manually, and shape descriptors were calculated similarly to in vitro platelet morphology assessments.

**Confocal IVM of the mesentery.** After induction of narcosis, animals received 15 μg of X488 antibody (Emfret) and 80 μg of fibrinogen-AlexaFluor 594 conjugate to visualize platelets and fibrin(ogen), respectively, and confocal microscopy of the mesentery was performed[32]. Briefly, the skin and peritoneum were opened along the midline, and 200 μl of warm PBS was pipetted into the abdominal cavity. Next, the proximal bowel was carefully exteriorized onto a glass coverslip and the mesenteric vessels exposed. Wet tissue paper was used to stabilize the bowel and prevent artifacts from bowel movements. A $1 \times 1$ mm piece of filter paper was saturated with 10% $FeCl_3$ solution and carefully placed in direct contact with a mesenteric vein for 3 min. Next, the ensuing thrombus formation and thrombus contraction was imaged on an inverted Zeiss LSM 880 in AiryScan Fast Mode ($\times 20/0.8$ obj., 990 ms/frame, laser power: 0.96%).

**$FeCl_3$-induced thrombus formation in the mouse carotid artery.** Arterial thrombosis was performed similar to published protocols[81]: mice were anesthetized, and the left carotid artery was exposed. A DyLight 488-conjugated Gp1b antibody (X488, Emfret, 50 μl) was injected via the tail vein to visualize platelets. Next, a filter paper (mm$^3$) was saturated with $FeCl_3$ solution (Sigma, 10%) and was placed abounding the vessel laterally. The paper was removed after 4 min and the ensuing thrombus formation was imaged on a fluorescence microscope (Axio-Scope; Carl Zeiss) with shots recorded every 5 min for a duration of 30 min. For image acquisition and analysis, a computer (Dell) with Cell$^R$ software (Olympus) was used. The platelet signal was used to quantify thrombus area in relation to vessel area over time. For histology, thrombi were retrieved after vessel occlusion and snap-frozen. Kryo-histology slices (10 μm thickness) were labeled with fibrinogen and CD41 antibody, and imaged on an LSM 880 confocal microscope with Airyscan module from Carl Zeiss ($\times 40/1.3$ oil immers. obj.).

**Deep vein thrombosis model of the IVC.** Thrombosis by ligation of the inferior vena cava (IVC) was conducted by performing a median laparotomy, and placing a space holder followed by a narrowing ligature[82], see Supplementary Fig. 1i. At 24 h after induction, the thrombus was retrieved and snap-frozen. Kryo-histology slices (10 μm thickness) were labeled with fibrinogen, CD41 antibody, and DAPI (4′,6-diamidino-2-phenylindole), and imaged on a Zeiss Axio Imager M2 epifluorescent microscope with a $\times 10$ and $\times 40/0.75$ air objective and an AxioCam MRm camera (Carl Zeiss Microscopy).

**Tail bleeding assay.** Mice were anesthetized as described above, 5 mm of the tail was cut with a precision scissor (MST) and the tail was immediately placed in 37 °C warm NaCl (0.9%) solution. Bleeding and re-bleeding times were recorded for 20 min.

**Lung infection with *S. aureus*.** For these experiments, we used bone marrow chimeric mice reconstituted with *PF4cre/Arpc2 Cre+/−* (KO) or *Cre−/−* (Ctrl) bone marrow. Anesthetized mice were intranasally injected with $3.2 \times 10^7$ CFU/g mouse weight[50] of *S. aureus* USA300. At 24 and 48 h p.i., 100 μl blood was drawn from the facial vein under brief anesthesia (isoflurane) and plated on BHI plates (Biomerieux) to assess bacteremia. At 48 h p.i., animals were sacrificed, and organs were carefully explanted under sterile conditions to avoid external contamination.

*CFU counts.* To assess CFU counts, blood, left kidney, left lung, and anterior spleen were weighted, and processed with a tissue homogenizer (4000 r.p.m., 30 s, Bead-Bug Homogenizer, Biozym). Then, serial dilutions were plated on BHI plates (Biomerieux) and colony counted after incubation for 20 h at 37 °C.

*Histology.* Organs were placed in 4% PFA for 1 h, 30% sucrose overnight, and then cryo-embedded. Ten-μm-thick lung cryo-slices were labeled with Gp1b-DyLight 649, *S. aureus* antibody-AlexaFluor 555 (secondary), Gr-1-AlexaFluor 488/anti-fibrinogen 488 (secondary), and anti-citrullinated H3 antibody (for quantification of NETosis and NETosis-primed neutrophils). We identified areas of high bacteria and neutrophil content classified as "abscesses." Areas of disseminated bacteria without neutrophil accumulation were classified as "infected tissue." Platelet–bacteria interaction was quantified in "infected tissue." A Zeiss Axio Imager M2 epifluorescent microscope with a $\times 40/0.75$ air objective and an AxioCam MRm camera (Carl Zeiss) or an LSM 880 confocal microscope with Airyscan module (Carl Zeiss, $\times 20/0.8$ air obj.) were used for image acquisition.

**Chimera generation.** Bone marrow chimeras were generated by irradiation of recipient C57Bl6 mice. Six- to eight-week-old mice were irradiated twice within a 4-h interval (195 kV, 10 mA) and received isolated bone marrow from the,

respectively, indicated mouse strains[83]. Bone marrow from cre-negative/WT littermates was used in the control group.

**LPS-induced lung injury model.** Two different models of acute lung injury were performed. Subacute lung injury (bleeding assessment)[41]: mice were anesthetized and received 20 μg of LPS (O111:B4) intranasally. After 24 h, animals were sacrificed, and BAL was obtained by flushing the lung twice with 1 ml PBS containing 2 mM EDTA. Hemoglobin content was assessed after sonication by fluorescence absorption (405 nm, Tecan Infinite F200), and flow cytometry was performed utilizing Rainbow Beads for cell content per volume quantification. For histology, lungs were directly harvested after 8 h, placed in 4% PFA for 1 h, and then in 30% sucrose overnight and cryo-embedded. Ten-micrometer slices were then stained with TER119-PE. Acute severe lung injury: to assess effects on survival, mice were anesthetized, and the trachea was exposed surgically[81]. Fifty micrograms of LPS (O111:B4) were injected directly into the lower trachea and bronchi. Anesthesia was maintained, for 4 h. Blood oxygen saturation was measured repeatedly by noninvasive pulse oximetry using MouseOx Plus (Starr Life Sciences, USA). The experiments were carried out in deep anesthesia as described above and were terminated (and counted as deceased), if mice showed clinical signs of respiratory failure or oxygen saturation below 60%.

**Bone marrow whole mount and flow cytometry.** To assess the effect of busulfan depletion on bone marrow function and ascertain time-dependent specificity, bone marrow from control and busulfan-treated WT animals was harvested. For whole mounts, bones were snap-frozen after PFA fixation (4%, 60 min) and sucrose treatment (30%, overnight). Bones were then cut longitudinally with a cryotome to expose marrow, and were stained with CD41-488, Gr-1-647, and TER119-PE (1:100) and imaged with a Zeiss LSM 880 confocal microscope ($\times 20/0.8$ obj.) in airyscan mode.

**Blood counts.** If not stated otherwise, blood counts were obtained in EDTA anticoagulated blood with an ABX Micros ES60 (Horiba Diagnostics) cell counter.

**Platelet isolation.** Human blood was drawn from the cubital vein into a syringe containing 1/7th volume of acid-citrate-dextrose (39 mM citric acid, 75 mM sodium citrate, 135 mM dextrose; ACD), diluted 1:1 with modified Tyrode's buffer (137 mM NaCl, 2.8 mM KCl, 12 mM $NaHCO_3$, 5.5 mM glucose, 10 mM HEPES, pH = 6.5), and centrifuged with $70 \times g$ for 35 min at room temperature (RT) with the break switched off. The supernatant consists of platelet-rich plasma (PRP), either directly used for experiments or used for the preparation of WPs by a 1:3 dilution in modified Tyrode's buffer (pH = 6.5) with PGI2 (0.1 mg/ml) and a second centrifugation step ($1200 \times g$, 10 min, RT) and re-suspension of the cell pellet in Tyrode's buffer (pH = 7.2). Platelet-poor plasma (PPP) was obtained by high-speed centrifugation of PRP ($14,000 \times g$, 5 min). Mouse blood was drawn intracardially from anesthetized mice and processed as described for human platelets. Baboon blood was drawn from a central venous catheter and processed similarly.

**2D migration/retraction assay.** For all in vitro migration/retraction experiments, human platelets were used if not indicated otherwise. Coverslips (no. 1.5, D263T, Nexterion) were acid washed (20% $HNO_3$) for 1 h and rinsed in ddH$_2$O for another hour. Coverslips were then air-dried and silanized with HMDS by spin coating at 80 r.p.s. (revolutions per second) for 30 s. Plastic channels (sticky slides IV$^{0.4}$, Ibidi) were attached in all experiments, except Supplementary Fig. 2d; here, a ring was glued to the coverslip to increase DIC imaging quality (open chamber assay).

*Fibrin(ogen) surface generation.* For fibrinogen surfaces, coverslips were coated with 37.5 μg/ml AlexaFluor-conjugated fibrinogen and 0.2% HSA in modified Tyrode's buffer for 15 min at RT. (Monomeric) fibrin surfaces were generated by the addition of 1 U/ml thrombin and 1 mM calcium. To create cross-linked fibrin substrates, 1:4 PPP or fibrinogen/HSA at concentrations resembling PPP concentrations (final concentration 625 μg/ml and 10 mg/ml, respectively) was added prior to thrombin addition. Visualization was ensured by spiking in labeled fibrinogen at 37.5 μg/ml.

*Fibrinogen edge.* Non-homogenous fibrinogen for haptotaxis experiments was coated by performing 1 μl dot coating for 15 min before attaching plastic channels. Then, washing and plating steps were performed as described above.

*Integrin ligands of tunable mechanical stability.* Acid-washed glass coverslips (HCO$_3$ (20%); 1 h at RT) were plasma cleaned (Zepto; Diener Electronic GmbH). Plastic channels ((sticky slides IV0.4, Ibidi) were attached to the coverslips and subsequently coated (30 min at RT) with PLL(20)-g[3.5]-PEG(2)/PEG(3.4)-RGD (1 mg/ml in PBS, SuSoS) to generate stabile, covalently bound RGD-containing substrates. Fragile RGD substrates were generated by introducing a biotin-neutravidin–biotin bridge linking RGD to the PLL–PEG (poly-L-lysine-PEG) backbone. The biotin–avidin bond ruptures at forces >160pN[84], thus allowing platelets to mechanically remove an RGD-containing ligand when integrin pulling

forces overcome 160 pN. Plasma-cleaned glass coverslips were coated with PLL (20)-g[3.5]–PEG(2)/PEG(3.4)–biotin (50%) (0.1 mg/ml diluted in 1 mg/ml PLL (20)-g[3.5]–PEG(2)/PEG(3.4) in PBS; 30 min at RT; SuSoS). PLL-g-PEG-biotin-functionalized coverslips were subsequently incubated with NeutrAvidin-FITC (200 μg/ml in PBS, Invitrogen) for 30 min at RT, followed by the incubation with cyclo RGDfk-biotin [Arg-Gly-Asp-D-Phe-Lys(Biotin-PEG-PEG)] (0.1 μM in ddH$_2$O, Peptides International) for 30 min at RT. Slides were washed with PBS and incubated with washed platelets in the presence of platelet activators (4 μM ADP, 2 μM U46619) and 1 mM Ca$^{2+}$/Mg$^{2+}$. Time-lapse movies were recorded as described above.

*Cell activation and plating.* Washed platelets were added to modified Tyrode's buffer containing 0.1% HSA (migration buffer) to reach a final concentration of $10 \times 10^6$ platelets/ml, activated by the addition of 4 μM ADP, 2 μM U46619, and 200 μM calcium, and pipetted unto the coverslips. After 15 min at 37 °C, non-adherent cells were removed by replacing buffer with cell-free migration buffer, if not indicated otherwise. After 45 min at 37 °C, cells were fixated with 1% PFA.

*Pharmacological inhibition studies.* For inhibition studies, cells were incubated for 20 min with CK666, CK689, Blb(−), and Blb(+) compounds. In some experiments, cell-free migration buffer containing inhibitors was superfused onto adherent cells. Plasmin treatment of surfaces. Fibrin-cross-linked surfaces were seeded with platelets and recorded for 15 min. Then, the platelet solution was replaced by modified Tyrode's buffer, and supplemented with calcium and plasmin (2.5 μg/ml) or H$_2$O (control group). Platelet behavior was recorded over time (45 min).

*Immunostaining.* For immunofluorescence studies, cells were fixated with PFA-GDA (2%/0.05%) for 20 min, and supplemented with NP40 (0.04%) for permeabilization. Primary and secondary antibodies were subsequently added in PBS buffer containing 1% BSA for 45 min. Antibodies included CD41 (1:100) Arp2/3 complex (1:200), and pMLC (1:200). Secondary antibodies and phalloidin–AlexaFluor 488 were added at a concentration of 1:200. PAC-1 antibody (1:50) was added directly to migrating/retracting platelets prior to fixation. Cells were assessed with an LSM 880 confocal microscope (Carl Zeiss) in Airyscan mode (40/1.3 oil immers. obj.).

*Time-lapse video microscopy.* Differential interference contrast (DIC), phase-contrast, and epifluorescence movies (20 s/frame) were recorded on an automated inverted Olympus IX83 microscope with an ×40/1.0 or ×100/1.4 oil-immersion objective. The microscope was equipped with a stage incubator (37 °C, humidified, Tokai Hit). For actin polymerization dynamics, LifeAct-eGFP platelets were imaged with a frame rate of 6 s.

*Plasmin treatment of surfaces.* Fibrin-cross-linked surfaces were seeded with platelets and recorded for 15 min. Then, the platelet solution was replaced by modified Tyrode's buffer, and supplemented with calcium and plasmin (2.5 μg/ml) or H$_2$O (control group). Platelet behavior was recorded over time (45 min).

*Biofilm assessment.* Coverslips were coated with monomeric fibrin (see Fibrin(ogen) surface generation) for 15 min. Fibrin surfaces were incubated with MRSA (1.4 × 10$^8$ CFU/ml) labeled with CFSE (5 μM) for 30 min at 37 °C and rinsed thoroughly to remove non-adherent bacteria. Cells were activated, plated, and fixated as described. For assessment of platelet recruitment, platelets were superfused on either fibrinogen-coated slices incubated with MRSA or buffer without the addition of ADP/U46619. Bacterial viability was tested after 45 minutes by adding LIVE/DEAD BacLight Bacterial Viability Kit (1.5 μl/ml) 15 min before fixation

**3D clot retraction**. 3D clot retraction was assessed by adding AF488–fibrinogen conjugate (15 μg/ml), Rhodamine 6G (0.0005%). and thrombin (1 U/ml) to PRP, and clot retraction was imaged after 20 min on a LSM 880 confocal microscope in AirySan mode (×20/0.8 obj., 2× zoom for overviews, and 5× zoom for detailed views, laser power: 0.96%)[85]. For inhibition experiments, the respective inhibitor was added 20 min prior to the experiment.

**In vitro platelet function assays**
*Activation flow cytometry.* Washed platelets ($2.5 \times 10^5$/μl) were diluted 1:24 in modified Tyrode's buffer, and activated by the addition of calcium (1 mM), and the stated compounds[86]. For inhibition experiments, cells were preincubated with the respective inhibitors for 20 min. At 5 min after activation, CD62P-APC and JonA-PE were added. After 20 min, platelets were fixated by adding 1:4 cold PFA 4%. Measurement was carried out on a BD Fortessa flow cytometer and analysis was completed with FlowJo (V10).

*Adhesion (in vitro thrombosis formation).* Whole blood was incubated with Rhodamine 6G to label platelets, recalcified, and perfused over a collagen-coated surface at the arterial shear rate (1000/s) for 5 min[87]. Rhodamine 6G-covered area was assessed by epifluorescence microscopy (Olympus IX83, ×40 objective).

*Aggregation.* Optical aggregometry was performed on PRP[81]. PRP was mixed with modified Tyrode's to obtain a platelet concentration of $1.5 \times 10^5$/μl. ADP (4 μM) or collagen (0.1 mg/ml) were added under continuous stirring at 1000 r.p.m. at 37 °C and measured in a two-channel aggregometer (ChronoLog 490-2D, Havertown, USA). The percentage of maximal platelet aggregation was analyzed 6 min after the addition of the agonist using Aggrolink software (ChronoLog, USA).

*Clot retraction.* PRP preincubated for 20 min with indicated inhibitors was recalcified, and 1 U/ml thrombin was added[88]. After 30 min incubation at 37 °C, clots were photographed and the remaining fluid quantified to assess the degree of retraction. As positive controls, we included cytochalasin D treatment completely abrogating platelet function by blocking actin polymerization, and myosin II blockade by blebbistatin(−).

**Western blotting**. Arp2/3 complex deletion was confirmed as described by Paul et al.[22], with minor variations. Obtained platelet pellets from Cre-positive and -negative *PF4cre-Arpc2fl/fl* mice were homogenized in RIPA buffer containing protease and phosphatase inhibitor cocktails (Thermo Fisher Scientific). Equal amounts of total protein (50 mg) were subjected to western blot analysis.

**Atomic force microscopy**. Atomic force microscopy (AFM) images were acquired according to the following protocol[89]: the mounted samples were attached to a glass slide fitting into the microscope stage with silica gel and imaged with an MFP-3D AFM (Asylum Research) and cantilevers of the MSNL Type (Type F, triangular, Bruker) at 512 × 512 pixels in tapping mode in liquid. Overview scans (10 × 10 μm$^2$) were acquired at a line rate of 0.25 Hz and representative 2 × 2 μm$^2$ regions were chosen for detailed views (line rate 0.5 Hz). Images were processed, and sections were analyzed with Asylum AFM software and IGOR Pro (WaveMetrics).

**Scanning electron microscopy**. For scanning electron microscopy (SEM), cells were incubated for 45 min in 0.1 M potassium-phosphate buffer containing 2.5% GDA, and then dehydrated with ethanol. Before sputter coating of the surface with gold (6 nm thickness, Cressington 108auto), a critical point dryer (Polaron E3000) was used along with standard drying protocol. Probes were imaged on an LEO DSM 982 SEM at 8 kV.

**Data analysis**. Data analysis was realized using FIJI[90] (ImageJ), Imaris (Bitplane), and ZenBlack (Zeiss). 4D in vivo data were stabilized with the ImageStabilizer plugin and dimensions were reduced by MIP. Cells were tracked with Manual tracking FIJI plugins. Tracks were further analyzed for velocity, directionality, and distance using Chemotaxis Tool (Ibidi) plugin. Motility pattern was classified in Rolling, defined as fast (>5 μm/min), directed movement in direction of flow, leukocyte-dependent motility for platelets showing displacement and direct contact with CD45+ leukocytes, adherence for platelets showing stable positioning over at least three video frames without discernible displacement on the vessel wall, and migration for repositioning of platelets on the vessel wall of more than one cell diameter, without direct contact to CD45+ leukocytes. Cells were tracked in their respective motility pattern for as long as they were visible in the recorded video. Aggregated platelets were excluded from the analysis. In vitro tracking and shape analysis of migrating and retracting platelets is described in detail elsewhere[16]. Briefly, cells were manually counted in phase-contrast/difference interference contrast, and cleared area was determined by using fluorescent fibrin(ogen) as substrate, and thresholding the cleared area in the respective fluorescent channel. For form analysis, cells were manually outlined, masked, and converted to binary images. For quantification, FIJI shape descriptors and the Celltool suite were used[91]. Distribution of subcellular proteins was assessed by drawing a line along the cell's shortest axis, defining front and rear. Resulting fluorescence profiles were parceled in ten (1–10, front to rear) segments, and MFIs were calculated for each region. Resulting MFIs were used to calculate averages across the cell population for each defined segment. For Arp2/3 distribution analysis, lines were placed adjacent to the pseudo-nucleus to avoid non-lamellipodial signal.

In biofilm experiments, bacteria size and percentage bundled by platelets were determined by thresholding fluorescently labeled bacteria (CFSE) and platelets (CD41-AF647), converting each channel to a binary image, and using FIJI ImageCalculator plugin to determine overlap. Next, shape and size descriptors were used to extract the parameters mentioned above. To assess bacterial viability, L/D-stained biofilms were thresholded in GFP (all bacteria) and RFP (dead bacteria) channels and percentages of non-viable bacteria were calculated. To determine the spatial distribution of adherent/migrating platelets in respect to neutrophil extravasation sites, we manually determined sites of neutrophil transmigration in our processed videos of the cremaster microvasculature. Next, we performed a SUM Projection of the time axis in the 649-channel visualizing platelets, defining areas of prolonged platelet contact. Using the interaction analysis plugin (MosaicIA), which performs an object-based interaction analysis in a spatial statistics framework based on nearest-neighbor distance distributions[92,93], we determined the spatial correlation strength between both patterns. Statistical significance was tested by F0 hypothesis testing. Colocalization of activated integrins and fibrin(ogen) substrate in in vitro experiments was visualized with FIJI plugin Coloc2. Data were post processed in Excel (Microsoft), Prism (GraphPad)

and Illustrator (Adobe). 3D surface rendering and pseudo-coloring was performed using Imaris (Bitplane).

**Statistical analysis and reproducibility.** Data are shown as means ± s.e.m. Statistical parameters, including the exact value of replicates ($n$) for individual experiments, can be found within the figure legends. We predefined groups for comparison prior to conducting the respective experiments. Representative data shown in the figures were reproduced at least in three independent experiments. For in vitro experiments, no prior statistical analyses to define sample sizes were used. Instead, we based sample sizes on previous experiments in our lab and others. Animal sample sizes were estimated using power calculations. To evaluate statistical differences between groups, $t$ tests, and analysis of variance (ANOVA) were performed after testing for a normal distribution ($F$ test or Kolmogorov–Smirnov test). If data did not fulfill the criteria of normal distribution, Mann–Whitney or Kruskal–Wallis tests were used, respectively. If ANOVA or Kruskal–Wallis tests showed statistically significant differences between groups, we performed post hoc tests (Tukey/Dunn's, respectively). In matched-sample experiments, the paired $t$ test was used. Only two-sided testing was applied. To analyze contingency table data, $\chi^2$ test was used. Survival was analyzed using the log-rank test. A $p$ value of <0.05 was considered statistically significant. Analyses were performed with Prism (GraphPad Software) and Excel (Microsoft). Age- and sex-matched mice were used for experiments. For adoptive transfer experiments, and hemorrhage assessment in the cremaster, the researcher performing the analysis and quantification of data was blinded to color allocation/genotype.

**Reporting summary.** Further information on research design is available in the Nature Research Reporting Summary linked to this article.

## Data availability
Data that support the findings of this study are available within the article and its Supplementary information. Source data for all figures and Supplementary figures are provided with the paper. Any additional information and related data are available upon reasonable request.

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

## Acknowledgements

We thank Sebastian Helmer, Nicole Blount, Christine Mann, and Beate Jantz for technical assistance; Hellen Ishikawa-Ankerhold for help and advice; Laura Machesky for providing PF4-Cre;Cyfip1fl/fl mice; Michael Sixt for critical discussions and the electron microscopy facility of IST Austria for excellent support. This study was supported by the DFG SFB 914 (S.M. [B02 and Z01], K.Sch. [B02], B.W. [A02 and Z03], C.A.R. [B03], C.S. [A10], J.P. [Gerok position]), the DFG SFB 1123 (S.M. [B06]), the DFG FOR 2033 (S.M. and F.G.), the DFG (M.B. [BE5084/3-2, TR240 project number 374031971]), the German Center for Cardio-vascular Research (DZHK) (Clinician Scientist Program [L.N.], MHA 1.4VD [S.M.], Postdoc Start-up Grant, 81×3600213 [F.G.]), FP7 program (project 260309, PRESTIGE [S.M.]), FöFoLe project 1015/1009 (L.N.), FöFoLe project 947 (F.G.), the Friedrich-Baur-Stiftung project 41/16 (F.G.), and LMUexcellence NFF (F.G.). This project has received funding from the European Research Council (ERC) under the European Union's Horizon 2020 research and innovation program (grant agreement no. 833440) (S.M.). F.G. received funding from the European Union's Horizon 2020 research and innovation program under the Marie Skłodowska-Curie grant agreement no. 747687.

## Author contributions

Initiation, F.G.; conceptualization, L.N. and F.G.; methodology, L.N., F.G., K.Sch., and M.L.; investigation, L.N., K.Sch., S.L., A.L., M.H., C.M., A.E., J.P., Z.Z., I.S., A-K.M., A.L., T.P., M.L., K.S., R.P., G.R., and V.Z; resources, L.W., B.U., C.A.R., B.W., C.S., M.B., S.X., R.L., and S.M.; formal analysis, L.N., A-K.M., A.L., K.P., and K.Sch.; writing—original draft, L.N. and F.G.; writing—editing, all authors; visualization, L.N. and F.G.; supervision, L.N., F.G., and S.M.; project administration, L.N. and F.G; funding acquisition, L.N., F.G., and S.M.

## Funding

## Competing interests

The authors declare no competing interests.
