## [Peer Review File · Nature Communications]

Reviewers' comments:

Reviewer #1 (Remarks to the Author):

Nicolai and colleagues study platelet haptotaxis in vitro on fibrin(ogen) and in vivo in a broad range of experiments using an inhibitor of the Arp2/3 complex and platelet-specific Arpc2-knockout mice.

In general, the huge variety of methodology used in this paper is outstanding, however often controls and in depth discussions are missing. The paper appears overloaded with data, some results are overstated and may not be really surprising, e.g. it has been shown before that lamellipodia are required for platelet migration, but not for thrombosis in vivo. With this background, results on the Arpc2 deficient mice and Arp2/3 inhibitor were expectable. The paper would benefit a lot focusing on the haptotactic migration with regard to its impact on inflammation. The microscopic image quality requires improvement and makes it difficult to draw definitive conclusions.

Analysis in general, less platelets are analyzed (e.g. minimum 25 – maximum 50 platelets of up to max 3 mice in total have been analyzed in Fig1). It is not clearly stated how many cells per mouse were analyzed in total, since only platelet numbers are provided (is 25 platelets 20 from mouse 1 and 5 from mouse 2 or 12+13)? Are platelet parameters between mice comparable? For correct statistical analysis, it is recommended to count platelets per mouse -> calculate mean of individual mice and then to calculate the mean for all mice (mean of means of individual mice). Please display the mean for all mice including standard deviation (not sem). At least three mice should be analyzed.

Figure 1, Supp 1 and video 1:

- Figure 1a/b: This finding is not surprising. No distinct migration behavior of single platelets is detectable from these images. Do these platelets get activated?
- The used magnification is very low making it impossible to identify and especially follow single platelets. The zoom-ins are of poor quality (not sharp and pixelated). Therefore, it is very difficult to conclude anything from the images/videos.
- How can the authors be sure that they are following the same platelet (e.g. during rolling) over the observation period? How can they discriminate between the same platelet in 2 frame rates and two different platelets in the 2 frame rates – as determined by their video frame rate settings? Using a different frame rate frequency would you get the same results?
A lot of movement in videos -> hard to follow single platelets, especially in part migration 1 and 2, it is impossible to judge whether the tracked platelet is still the same from frame rate to frame rate.
- Fig 1b/ S1: "Platelets either transiently or firmly adhered to the vessel wall and a fraction of adherent platelets subsequently migrated autonomously". For how long did the authors follow individual platelets for migratory movement? Did they calculate the mean velocity over the whole observation period or just of moving platelets? How did they handle transiently adhered platelets? How many platelets are firmly adhered and do not move at all? What is the % distribution of migrating vs. adhered vs. rolling vs. leukocyte-bound platelets?
- Analysis and images of non-inflamed control group is missing.
- 1c: Labelled fibrinogen was injected into mice to visualize fibrin(ogen) deposition. Results display a spotty deposition of fibrinogen here and there. Would an anti-fibrinogen antibody not be better suited to visualize total fibrinogen deposition, since the authors might miss a lot deposition of the endogenous, highly abundant, but unlabeled fibrinogen ?
- 1d: How many platelets (%) do show lamellipodia formation in this setting? Is this a frequent or rather rare event? Please add a fibrinogen staining here. One would assume that they form lamellipodia most probably in the presence of deposited fibrinogen.
- 1e: These results are not surprising as it has been shown before by the Bender lab (2019) that lamellipodia formation is not required for thrombus formation / stabilization.
- 1f; Supp 1h, i: Thrombus very densely packed, filopodia formation not visible due to the density and low resolution. Would be better to either show this by scanning electron microscopy or to

transfer a low number of labelled platelets for better visualization.

Overall, the paper would very much benefit from a life cell imaging actin dye or a photo-switchable/convertible dye like dendra2 for in vivo imaging of these cellular processes, as they are very difficult to judge from the provided images and videos.

Figure 2:

- Fig 2: How different are mouse (in vivo data Fig1) and human platelets (Fig 2) keeping in mind that GPVI responses to fibrinogen/fibrin are critically different.
- Fig 2e: Please indicate more precisely: Staining of fibrin pre- and post-plasmin treatment? How do platelets affect plasmin treatment? BF images at time = 0: should there not be fibrin fibers in control and plasmin condition and at time = 25 in control conditions? How would a platelet behave on a control coating (control protein), a plasmin only coating and an uncoated surface?
- Movie 2: plasmin part -> please provide a sharp video for the fibrin(ogen) – channel.

Figure 3:

- Since Arp2/3 deficient platelets are known to have a spreading (no lamellipodia formation) and integrin activation defect (Paul et al., Blood Adv 2017), the results on the Arp2/3 inhibitor and Ko mice are to be expected.
- Please provide a WB or IF staining confirming the absence of the protein from Arp2c-Ko platelets.
- 3j: In vivo thrombus formation on A. carotis does not lead to an occlusive thrombus formation in WT mice here (only 50% vessel occlusion). WT mice were treated with Busulfan to lower platelet count to Ko levels. Can you exclude an effect of the busulfan treatment on platelet function? Please compare untreated and Busulfan-treated WT platelets in terms of platelet reactivity and function. A platelet count of 50% mice should be enough for normal thrombus formation in vivo. Or was only a mild injury chosen and is the thrombus formation defect in vivo in Arpc2-knockout mice more prominent using a severe injury? 3-6 mice is clearly a (too) low number for an in vivo thrombosis experiment. Please indicate the distribution of the mice in your histogram.
- 3k: Why are the WT mice in tail bleeding time assay so variable? Please increase n-number.
- 3l: Please provide images/videos for kinetic studies of thrombus formation of mesenteric arterioles as well as a quantitative analysis of thrombus formation in vivo over time.
- Movie 5: s. comment on video acquisition and frame rate frequencies. The provided movie does contain stationary/adhered/non-migrating as well as moving Arp2c-ko platelets, more videos and extensive analysis are necessary.

Figure 4:

- In vivo experiments were stated to be performed with adjusted platelet count between WT and Ko platelets. Platelet/visual field are less in Ko mice than in WT (Fig 4a, e, f). Can you comment on this?

4f: this would also explain the reduced colocalization of platelets to low alpha-SMA areas.

- Platelet distance from endothelial junctions is reduced in KO mice. Are vessel size, vessel number, vessel diameter and endothelial junctions comparable between KO and WT mice? Is fibrinogen deposition on the vessel wall comparable between KO and WT mice? Are ECM proteins exposed as well and influence platelet haptotactic behavior?
- Image of Arp2c-KO mice displays an increased number of PMN in inflamed cremaster. Is this difference statistically significant or is this a non-representative image?
- 4g: How were the image analyzed for microbleedings? Mice were perfusion fixated to remove all intravascular RBCs. Did the authors do a complete stack of the cremaster muscle and a 3D reconstruction of all vessels to make sure that residual Ter119 staining is outside the vessel? This is not clear from the images. Does n represent images or animals?

Figure 5:

- Is fibrinogen deposition altered in the lung of KO mice?
- Is the amount of PMN-platelet conjugates comparable in KO and WT?
- Sup Fig 5g, h: Please provide platelet counts of mice after anti-hIL4-antibody mediated platelet depletion before and after the transfer of WT/KO platelets.
- 5d and Sup. Fig 5m: Values for Hgb in Bal more than 10 fold lower than in other Hgb measurements, but erythrocyte count in BAL is comparable. Do the authors have any explanation for this?

- Can you explain the different results in inflammatory lung bleeding after LPS treatment regarding a pharmacological inhibition of α IIb and genetic deletion?

Figure 6:

- Movie 6: are MRSA bacteria engulfed in platelets or do they bundle on to the surface?
- 6a: What does PH stand for?
- 6e: How biologically relevant is reduction from 3% to 1% of MRSA associated with platelets? At which time point after inoculation were d and e analyzed?

Minor comments:

- Fig 1b: How many mice were analyzed? Figure legends just indicate 39 platelets. Please check complete manuscript and discriminate between number of experiments and number of mice.
- 3j-l: WT mice were treated with busulfan to reduce the plt count to KO-levels. Please include this in the figure legend and figure labelling.
- Length of scale bar missing in figure legend 1 for all microscopic images.
- In general, due to the overload of data, explanations are kept too short or too general- especially regarding the supplemental figures (e.g. no explanation of Blebbistatin and cytochalasin treatment in supplemental Figure 4).

Reviewer #2 (Remarks to the Author):

In the present manuscript 'Vascular surveillance by haptotactic blood platelets in inflammation and infection', Nicolai and colleagues report a critical role of Arp2/3 complex-mediated lamellipodia formation in the sealing of vascular microinjuries post neutrophil extravasation (assessed upon LPS-mediated cremaster and lung inflammation) in a fibrinogen-dependent manner. In addition, the authors show that platelets prevent bacteria and erythrocyte dissemination in the context of infection. With this the authors report for the first time a role of migrating, haptotactic platelets in safeguarding inflamed vessels. Although the study is novel, extensive, well designed and controlled as well as of great interest to the broad readership of Nature Communications, there are some aspects that should be addressed.

Major comments:

1. How do the authors explain the absence of microbleeds/inflammation-associated bleeding in the lung of other mouse models incapable of forming platelet lamellipodia, e.g. *Cyfp1*^{-/-} (Schurr et al. Blood 2019). Similarly, why does the reverse passive Arthus reaction not cause bleeding in the skin of *Arpc2*^{-/-} (Paul Blood Adv 2018) and *Cyfp1*^{-/-} (Schurr et al. Blood 2019)? This should at least be discussed.

2. Due to the emerging evidence from single cell RNA sequencing experiments on different origins, molecular identities and functions of endothelial cells from different vascular beds would it be of value to additionally visualize platelet migration in another easily accessible vascular bed, e.g. mesentery/ear skin to exclude vascular bed-specific effects. Are the observed effects specific to LPS-mediated inflammation? What happens in response to a more generalized inflammation, e.g. in response to TPA? Moreover, proof of vascular integrity breakdown is missing, injection of fluorescent beads could help to demonstrate vascular leakage. This is specifically evident in Fig. 1c-d. In 1c junctions are clearly discontinuous, which is in agreement with vascular leak but not in 1d, where junctions still show a zipper-like structure.

3. The authors clearly show that platelet morphology and the underlying mechanisms differ between inflammation and thrombosis. This raises the question on whether platelet migration plays a role in thrombo-inflammatory processes post transient mid cerebral artery occlusion. In addition, what happens upon induction of minor endothelial damages e.g. by using less FeCl_3 (e.g. 7%) and/or reduced incubation time which does not lead to full vessel occlusion?

4. Nicolai and colleagues convincingly demonstrate that platelets migrate towards higher ligand

densities. However, based on the tracks presented in Fig. 3e it is tempting to speculate that additional mechanisms might contribute to platelet 'decision' making. How do the authors explain that all displayed platelets on the low density matrix (even the ones farther away) migrate towards the high density fibrinogen interface although they are surrounded by a homogenous fibrinogen matrix and thus should display a random and not directed migration?

5. Overall it is difficult to follow in which mouse model ('native' control and mutant *Arpc2* mice, platelet count-adjusted (antibody- or drug-mediated), bone marrow chimeras or platelet transfusion) the presented experiments were performed. Additional information on the model and explanation on why the model was chosen would help to improve the comprehensibility of the study. Depending on the used model, certain controls are missing, e.g. fibrinogen localization in *Arpc2*^{-/-} mutants. Unaltered fibrinogen distribution would strengthen the notion of impaired migration. Moreover, *Arpc2*^{-/-} pictures in Figure 4a, e, f do not reflect statistics presented in Figure 4c and Supplementary Figure 4h as all of them show less platelets for mutants than controls. Why is the adhesion of transfused but not native *Arpc2*^{-/-} platelets to inflamed cremaster vessel almost doubled (Fig. 4c; Supplementary Fig. 4h)? Is integrin trafficking also altered/ contributing to the observed defects? It would be interesting to assess integrin turnover and closure which were previously shown to be dependent on remodelling of the cortical actin cytoskeleton.

6. The presented pictures on the acute lung injury model (Fig. 5a,b,e) are not very informative. Inclusion of Cdh5/Cd31 staining would help to obtain some morphological information. It's difficult to see lamellipodia in the presented pictures. Pictures of untreated controls for comparison are missing. How did the authors distinguish between intact vasculature and bleeding spots for their quantifications? Again, beads could help to address this question.

7. Fig 6e: Why was this model performed in bone marrow chimeras? Were platelet counts equal for both groups?

Minor comments:

1. Line 104: "Supplementary Fig. b-d" should read as Supplementary Fig. 1b-d.

2. Line 250: "Supplementary Fig. 7h" should read as Supplementary Fig. 3h.

3. Fig 4h and Suppl. Fig. 4e, f: Why are there so many intravascular erythrocytes despite perfusion?

4. The statement 'Taken together, Arp2/3-dependent spreading and haptotaxis constitute crucial platelet functions required for safeguarding the inflamed microvasculature and preventing inflammatory bleeds.' seems slightly overstated considering the unaltered results of the rPA model in *Arpc2*^{-/-} animals (Paul Blood Adv 2018).

5. Fig. 6a: Is the biofilm clearance already correlated to the number of platelets or does it just reflect the increased number of recruited platelets and hence clearance in the presence of MRSA?

Reviewer #3 (Remarks to the Author):

The authors examine the role of Arp2/3 complex in regulating platelet shape and motility. They use an impressive array of in vitro and in vivo approaches in their study. Their findings reveal an important role for Arp2/3 in positioning platelets to microlesions in the inflamed microvasculature

and capture of pathogenic bacteria. This is a superb paper that sheds important light on platelet movement in vivo and the role of Arp2/3 in general. I have only minor comments for the authors consideration.

-Page 5. "we detected a thin layer of fibrin(ogen), providing a two-dimensional platform for adhesion of single platelets in vivo (Fig. 1c)". Can the authors comment further on the pattern of fibrin(ogen) deposition observed? Actual depth? Patterns? Structures?

-Page 6 "Importantly, cleavage of 3D fibrillar fibrin mesh-works by the serine-protease plasmin rescued polarization and migration, highlighting the ability of platelets to instantaneously adapt their motility program to the physical composition of the microenvironment (Fig. 2e-f and Supplementary Video 2)". These are interesting data but may reflect other effects of the enzyme. Have the authors tried an inactive (e.g. denatured) enzyme control?

-Figure 6g, The authors report CFUs/mL blood. The y-axis should be Log10 scale.

-Figure 6h. CFU data should be included for kidney and spleen for comparison with g.

-The authors may wish to discuss Wiscott Aldrich Syndrome, which pointed to Arp2/3 regulation in immune control.

Point-to-point response to Reviewer #1:

We thank this reviewer for his/her very thorough evaluation of our manuscript. Please find below our point-to-point response.

Reviewer #1 (Remarks to the Author):

Nicolai and colleagues study platelet haptotaxis *in vitro* on fibrin(ogen) and *in vivo* in a broad range of experiments using an inhibitor of the Arp2/3 complex and platelet-specific Arpc2-knockout mice. In general, the huge variety of methodology used in this paper is outstanding, however often controls and in-depth discussions are missing. The paper appears overloaded with data, some results are overstated and may not be really surprising, e.g. it has been shown before that lamellipodia are required for platelet migration, but not for thrombosis *in vivo*. With this background, results on the Arpc2 deficient mice and Arp2/3 inhibitor were expectable. The paper would benefit a lot focusing on the haptotactic migration with regard to its impact on inflammation. The microscopic image quality requires improvement and makes it difficult to draw definitive conclusions.

We thank Reviewer 1 for the positive evaluation of our manuscript and for his valuable comments. We appreciate that he/she highlights the multimodality approach we used to study the spatio-temporal regulation of platelet recruitment to sites of inflammation.

We apologize if we were unclear in pointing out the novelty of our findings. As stated correctly by this reviewer others have previously demonstrated that lamellipodia are dispensable for hemostatic and thrombotic platelet functions. We actually alluded to this fact in the introductory paragraph of the first version of our manuscript and cited relevant previous studies (see citations Paul et al. Blood Advances 2017 and Kahr et al. Nature communications 2017). Also, we adapted the introduction to specifically outline the novelty of our work. Another more recent study confirming that lamellipodia are not required for thrombosis and hemostasis is now also included in the revised version (see reference 23). Hence, several studies established that platelets do not require lamellipodia for thrombosis and hemostasis. Yet, whether platelet lamellipodia formation has any physiological role remains entirely unclear. Our study therefore is the first to show that platelets require Arp2/3 to form lamellipodia *in vivo* and that this is essential for the course of the inflammatory response. While we previously identified platelet migration as an important platelet function during inflammation and infection, the mechanistic role of lamellipodia and branched actin networks in this process as well as its *in vivo* relevance remained unclear. We therefore clearly disagree with the statement that the “results on Arpc2 deficient mice” in the context of platelet migration *in vivo* were “expectable”.

To address the reviewers concerns regarding the complexity of our data, we now added further explanation of the methodology within the results section and figure legends (See text and methods) in the revised manuscript (also see point-to-point response below).

We appreciate the reviewer's suggestion to streamline our data and to further emphasize the role of platelet haptotaxis in inflammation in the revised version of the manuscript. We therefore moved thrombosis/clot retraction/hemostasis experiments in Arpc2 -/- (Fig. 3 j-l) from the main figures to the supplement (Supplementary Figure 4 of the revised manuscript). While Figure 1 and 2 show important novel data contrasting platelet functions in thrombosis and hemostasis with inflammatory platelet

effector functions, main Figures 3-7 now exclusively focus on the mechanisms and *in vivo* relevance of platelet haptotaxis in platelet biology and inflammation. In addition, we shortened the introductory paragraph to focus more on the haptotactic migration with regard to its impact on inflammation (see revised version of the manuscript).

Analysis in general, less platelets are analyzed (e.g. minimum 25 – maximum 50 platelets of up to max 3 mice in total have been analyzed in Fig1). It is not clearly stated how many cells per mouse were analyzed in total, since only platelet numbers are provided (is 25 platelets 20 from mouse 1 and 5 from mouse 2 or 12+13)? Are platelet parameters between mice comparable? For correct statistical analysis, it is recommended to count platelets per mouse -> calculate mean of individual mice and then to calculate the mean for all mice (mean of means of individual mice). Please display the mean for all mice including standard deviation (not sem). At least three mice should be analyzed.

We thank the reviewer for his/her comment. We now corrected all Figure legends and clearly state how many cells were pooled from how many mice. Additional experiments were performed to ensure that at least three mice were analyzed. The new data is now included in the revised version of the manuscript (please see Figure 4 (cremaster adoptive transfer experiment and analysis of platelet positioning regarding VE-Cadherin/alpha SMA) and Supplementary Figure 1 (clot retraction analysis).

To show that platelet parameters are comparable between mice we now display the mean for all mice including standard deviation (please see Reviewer Figure 1 a-d).

Figure 1, Supp 1 and video 1:

- Figure 1a/b: This finding is not surprising. No distinct migration behavior of single platelets is detectable from these images. Do these platelets get activated?

We thank the reviewer for this valuable comment and improved the tracking lines in Fig 1b to highlight the path of platelet migration. However, we want to emphasize that platelets have not been shown to migrate in the cremaster vasculature following LPS mediated inflammation and we therefore disagree with the statement that Figure 1b is “not surprising”.

Several lines of evidence indicate that platelets are activated during haptotaxis *in vivo*: (1) Previous reports have shown that platelets becoming adherent in the (micro-) vasculature need to get activated to engage their adhesion receptors (Offermanns et al, *Circ Res* 2006, Nesbitt et al, *Nature medicine*, 2009). (2) In line with this, we show here that platelet haptotaxis within the microvasculature depends on integrin $\alpha_{IIb}\beta_3$ (Supplementary Figs. 5j and 6m). (3) In addition, we have previously shown that Integrin $\alpha_{IIb}\beta_3$ needs to be activated to engage Fibrin(ogen) and to promote platelet migration, a process dependent on platelet activation (Gaertner et al. *Cell* 2017).

- The used magnification is very low making it impossible to identify and especially follow single platelets. The zoom-ins are of poor quality (not sharp and pixelated). Therefore, it is very difficult to conclude anything from the images/videos.

We thank the reviewer for this very valuable comment. We apologize that we had to compress images and videos for the initial submission of the manuscript causing poor resolution. We now include high

definition images to improve this in the revised version. Since platelets are of very small size (1-2 μm) forming submicron protrusions, we agree that the intravital imaging performed in our study is at the resolution limit of state-of-the-art intravital-imaging-setups. Yet, we established an imaging protocol that reliably allows us to visualize and analyze single platelets during inflammation (see Gaertner et al, *Cell*, 2017 and also see detailed response to your next comment). To further validate if our imaging setup can reliably detect impaired platelet migration in the cremaster microvasculature, we now performed an additional control experiment using MYH9^{-/-} platelets that cannot generate force and are therefore limited in their migratory capacity (Gaertner et al, *Cell*, 2017). Indeed, MyH9^{-/-} platelets showed a similar motility pattern in the inflamed cremaster microvasculature as compared to Arpc2^{-/-} platelets, confirming that our imaging and analysis protocol can reliably detect impaired migration of platelets (Reviewer 1 Figure 2 a). In contrast to Arpc2^{-/-} platelets, Myh9 deficient platelets however were unable to retract clots (Reviewer 1 Figure 2 b), highlighting the higher specificity of the Arpc2^{-/-} mouse-model for studying platelet migration *in vivo*.

- How can the authors be sure that they are following the same platelet (e.g. during rolling) over the observation period? How can they discriminate between the same platelet in 2 frame rates and two different platelets in the 2 frame rates – as determined by their video frame rate settings? Using a different frame rate frequency would you get the same results? A lot of movement in videos -> hard to follow single platelets, especially in part migration 1 and 2, it is impossible to judge whether the tracked platelet is still the same from frame rate to frame rate.

We are grateful for this in-depth assessment of our setup and for raising these important questions. We used spinning-disc intravital microscopy, which is the state-of-the-art imaging technique to record confocal stacks at high frame rates. We imaged whole microvessels (avg 20-25 μm in depth) with high z-resolution (z-step: 2.5 μm) at high speed to achieve a frame rate of 20 seconds. Laser power was adjusted to a minimum to avoid phototoxicity and photobleaching (30% of max). Based on our previous studies (Gaertner et al. *Cell* 2017) we know that platelets migrate at rather low speeds (1-5 $\mu\text{m}/\text{min}$) both *in vitro* and *in vivo*. Consequently, at the chosen frame-rate, migration distances from one frame to the other are below the mean platelet diameter of 1-2 μm and thus platelets contours from one frame to the other are overlapping. Thus, based on these settings we were able to (1) identify and (2) track a single platelet over time. In contrast, platelet rolling is significantly faster (100 $\mu\text{m}/\text{min}$). We agree that due to these significantly higher speeds the chosen imaging parameters, which are optimized for studying platelet migration are at the limit of the required temporal resolution. However, the values we obtained are in the range of a previous study where we used Confetti-color-coding to ensure that the same platelet was tracked over time (Gaertner et al. *Cell* 2017).

- Fig 1b/ S1: “Platelets either transiently or firmly adhered to the vessel wall and a fraction of adherent platelets subsequently migrated autonomously”. For how long did the authors follow individual platelets for migratory movement?

Our average recording time was ~35 minutes (see Reviewer 1 Figure 2c). This is now mentioned in the Material and Methods section of the revised manuscript.

Did they calculate the mean velocity over the whole observation period or just of moving platelets? How did they handle transiently adhered platelets?

This is an important question. Migration (both *in vivo* and *in vitro*) was defined by platelet displacement of at least one cell diameter. When this criterion was fulfilled, platelets were tracked from frame-to-frame and the mean velocity of the whole track was calculated (track-length (um) divided by imaging time (min)). Transiently adherent platelets counted as rolling cells if they adhered for less than <3 frames. We extended the methods to explain the tracking strategy in greater detail (see revised Methods).

How many platelets are firmly adhered and do not move at all? What is the % distribution of migrating vs. adhered vs. rolling vs. leukocyte-bound platelets?

We are grateful for this valuable question. Based on our data shown in Figure 4c of the manuscript we calculated the following fractions: 20.99(±3.28)% for adhesion, 35.23(±4.63)% leukocyte bound motility, 11.59(±2.41)% platelet migration and 32.18(±3.74)% rolling platelets.

- Analysis and images of non-inflamed control group is missing.

As suggested, we now performed “non-inflamed” control experiments by injecting NaCl (see Reviewer 1 Figure 2d). In contrast to LPS-injection, we did not detect fibrin(ogen) deposition, platelet or neutrophil recruitment. This information is provided in the revised manuscript (see Supplementary Fig. 1a).

- 1c: Labelled fibrinogen was injected into mice to visualize fibrin(ogen) deposition. Results display a spotty deposition of fibrinogen here and there. Would an anti-fibrinogen antibody not be better suited to visualize total fibrinogen deposition, since the authors might miss a lot deposition of the endogenous, highly abundant, but unlabeled fibrinogen?

Thank you very much for this valuable thought. To address the question, whether plasma spiked with fluorescently labeled fibrinogen allows to reliably detect fibrin formation, we mixed labeled fibrinogen (Sigma, Fibrinogen conjugated with AlexaFluor 488 dye) in a 1:20 relation to plasma fibrinogen (comparable to the dilution used *in vivo*, see Reviewer 1 Figure 2e) and generated fibrin *in vitro* (also see Methods section: 2D migration/retraction assay). We then stained the fibrin-network with an anti-Fibrinogen antibody (Alexa 647). Indeed, both stainings highly overlapped verifying the homogenous integration of labeled fibrinogen into the fibrin network.

- 1d: How many platelets (%) do show lamellipodia formation in this setting? Is this a frequent or rather rare event? Please add a fibrinogen staining here. One would assume that they form lamellipodia most probably in the presence of deposited fibrinogen.

54.54% of platelets recruited to sites of leukocyte extravasation in the inflamed microvasculature showed a significantly larger projected area compared to platelets recruited to thrombi, indicating lamellipodia formation and spreading (see Reviewer 1 Figure 2f,). Consequently, platelets frequently spread at sites of leukocyte extravasation. A fibrin(ogen) co-staining is shown in Figure 1c.

- 1e: These results are not surprising as it has been shown before by the Bender lab (2019) that lamellipodia formation is not required for thrombus formation / stabilization.

We thank the reviewer for this comment and now cited Schurr et al. Blood 2019 in the revised version of the manuscript (see REF 23). While the Bender lab demonstrated by scanning electron microscopy that platelets do not form lamellipodia at the shell region of arterial thrombi of the aorta, platelet morphology was not yet assessed in the cremaster microvasculature. Since we aimed to directly compare lamellipodia formation following thrombotic and inflammatory stimuli it was essential (1) to perform both types of injuries within the same vascular bed, (2) to use the same sample fixation protocol and (3) to analyze the samples with the same imaging setup. We therefore believe that it is crucial for the interpretation of our results to also show the analysis of platelet morphology within microthrombi in the cremaster vasculature.

- 1f; Supp 1h, i: Thrombus very densely packed, filopodia formation not visible due to the density and low resolution. Would be better to either show this by scanning electron microscopy or to transfer a low number of labelled platelets for better visualization. Overall, the paper would very much benefit from a life cell imaging actin dye or a photo-switchable/convertable dye like dendra2 for *in vivo* imaging of these cellular processes, as they are very difficult to judge from the provided images and videos.

We thank the reviewer for this comment. We used PF4cre +/-; RS26-Confetti^{fl/fl} mice to be able to delineate individual platelets in a densely packed, growing thrombus (Fig 1e and Suppl Fig 1h). We agree that visualization of thin, submicron-sized filopodia *in vivo* is challenging and may require additional imaging strategies (such as life cell imaging actin dye or a photo-switchable/convertable dye like dendra2). While studying filopodia within thrombi was not the major aim of our study, we now performed additional experiments to provide further evidence of filopodia formation without lamellipodia formation in platelets recruited to thrombi *in vivo* by live-imaging (see time series of mesenteric thrombus formation in Reviewer 1 Figure 3a). Importantly, these results are well in line with a recently published study using scanning electron microscopy to show that platelets do not form lamellipodia but rather filopodia during thrombus formation (Schurr et al. Blood 2019).

Figure 2:

- Fig 2: How different are mouse (in vivo data Fig1) and human platelets (Fig 2) keeping in mind that GPIIb/IIIa responses to fibrinogen/fibrin are critically different.

We did not detect major differences between human and mouse platelets (See Suppl. Figure 3j and Fig. 6). Platelets of both species adapt to the mechanical properties of fibrin(ogen) deposits: on cross-linked

substrates, all platelets showed retraction behavior, whereas fibrinogen coated substrates triggered efficient migration. Interestingly, we observed this conserved response to substrate mechanical properties also in baboon platelets (See Reviewer 1 Figure 3b). To address if GPVI plays a role in this process we incubated mouse platelets with the GPVI blocking antibody (JAQ1). This treatment did not change their migratory behavior (see Reviewer 1 Figure 3c).

- Fig 2e: Please indicate more precisely: Staining of fibrin pre- and post-plasmin treatment?

We used pre-labeled fibrinogen (Fibrinogen-AlexaFluor488, Sigma), this has now been indicated more precisely in the methods part.

How do platelets affect plasmin treatment?

In our hands, platelets did not affect the dissolving of fibrin fibers in a relevant way, as visualized and analyzed in Reviewer 1 Figure 3d.

BF images at time = 0: should there not be fibrin fibers in control and plasmin condition and at time = 25 in control conditions?

Thank you for this very valuable observation. Indeed, under control and plasmin conditions, fibers are present at time = 0 and then gradually degrade in plasmin conditions (also see Movie 2). Large fibers are not evenly distributed and not all are visible in phase contrast. However, we replaced the image by a more representative cell where fibers are visible in phase contrast images at time=0.

How would a platelet behave on a control coating (control protein), a plasmin only coating and an uncoated surface?

We apologize for the misunderstanding. We did not use plasmin coated surfaces – we used cross linked substrates, and reversed cross-linking by adding plasmin (see scheme Fig. 2e of the rev. manuscript). In addition we performed a control experiment with heat-inactivated plasmin (Reviewer 1 Figure 3e).

- Movie 2: plasmin part -> please provide a sharp video for the fibrin(ogen) – channel.

As one can see in the Phase contrast micrograph, image focus is not the problem leading to decreased image quality in the fibrin(ogen) channel – the addition of plasmin partly cleaves the AlexaFluor488 fluorescence tag, leading to loss of fluorescent signal. We now corrected the loss of fluorescence by normalizing the images of the movie to the same mean intensity using the “bleach correction” plugin in FIJI (Fiji Is Just Imagej), see Supplementary Video 2.

Figure 3:

- Since Arp2/3 deficient platelets are known to have a spreading (no lamellipodia formation) and integrin activation defect (Paul et al., Blood Adv 2017), the results on the Arp2/3 inhibitor and Ko mice are to be expected.

Paul et al. convincingly demonstrated that deletion of the Arp2/3 complex using *PF4-Cre; Arpc^{fl/fl}* specifically abrogates platelet lamellipodia formation while only mildly affecting integrin activation and granule secretion, thus providing an ideal model to study the role of lamellipodia in platelet physiology. We apologize for not citing this important article in this section of our manuscript and now added the citation at pg. 10 line 9. While Paul et al. investigated a large variety of platelet functions in *PF4-Cre; Arpc^{fl/fl}* mice and concluded that Arp2/3 deletion does not cause a significant difference in hemostasis and thrombosis compared to wildtype controls, we here (Fig. 3) investigated haptotaxis as a novel yet undescribed platelet function. Hence, the mechanisms of platelet haptotaxis were unknown and a role of Arp2/3 in this process remained to be determined. Consequently, we do not agree that our results on Arp2/3 inhibition in the context of haptotaxis and migration were to be expected and apologize, if Figure 3 was not clearly showing the novelty of our findings. To streamline and emphasize our novel findings in Figure 3, we now moved panels 3j-l, showing data confirming unaffected thrombosis and hemostasis in *PF4-Cre; Arpc^{fl/fl}* mice to Supplementary Fig 4.

- Please provide a WB or IF staining confirming the absence of the protein from Arp2c-Ko platelets.

We included a Western Blot in Reviewer 1 Fig. 4 a.

- 3j: In vivo thrombus formation on A. carotis does not lead to an occlusive thrombus formation in WT mice here (only 50% vessel occlusion). WT mice were treated with Busulfan to lower platelet count to Ko levels. Can you exclude an effect of the busulfan treatment on platelet function? Please compare untreated and Busulfan-treated WT platelets in terms of platelet reactivity and function.

We are grateful for this important comment. To assess platelet reactivity and function after Busulfan treatment we now performed additional control experiments. WT mice were either injected with Busulfan or NaCl, and platelets were isolated after 14 days according to our protocol. Extensive in vitro comparison showed comparable integrin $\alpha_{IIb}\beta_3$ activation upon stimulation with ADP, thrombin and collagen (Reviewer 1 Figure 4 b). No differences in clot retraction (Reviewer 1 Figure 4 c), and importantly, comparable morphology and migration behavior (Reviewer 1 Figure 4 d-e) were observed.

A platelet count of 50% mice should be enough for normal thrombus formation in vivo. Or was only a mild injury chosen and is the thrombus formation defect in vivo in Arpc2-knockout mice more prominent using a severe injury?

3-6 mice is clearly a (too) low number for an in vivo thrombosis experiment. Please indicate the distribution of the mice in your histogram.

Thank you for this comment. For an arterial thrombosis model with complete occlusion in the genetic model of Arpc2 deficiency in platelets, we would like to refer you to Paul et al, *Blood Advances* 2017. Here the authors did not see a difference in clot formation, highlighting that there is no role of Arp2/3 in thrombus formation. We repeated a thrombosis model to confirm their results but decided to use a milder injury. In our experience (Pircher et al, *Nat Comm* 2017, Marx et al, *Blood* 2019), increased FeCl injury leaves no room to detect minor changes due to extensive damage. Also, thank you for your comment on the mouse numbers and distribution. We actually performed the experiment on n=5-6

mice and apologize for not stating this correctly. You find a detailed distribution of the mice in Reviewer 1 Figure 5a.

- 3k: Why are the WT mice in tail bleeding time assay so variable? Please increase n-number.

Thank you for this comment. We agree that the data of the tail bleeding assay was variable and now increased the n-number to n=7 (see Suppl. Fig 4c of the revised manuscript). Our findings confirm data by Paul et al. *Blood Adv* 2017, as this group also did not see a difference in tail bleeding times in conditional Arpc2 KO mice compared to control.

- 3l: Please provide images/videos for kinetic studies of thrombus formation of mesenteric arterioles as well as a quantitative analysis of thrombus formation in vivo over time.

Thank you for this valuable comment. We now performed a detailed kinetic analysis of thrombus formation in mesenteric arterioles (see Reviewer 1 Figure 4b).

- Movie 5: s. comment on video acquisition and frame rate frequencies. The provided movie does contain stationary/adhered/non-migrating as well as moving Arpc2c-ko platelets, more videos and extensive analysis are necessary.

Thank you for this comment. Video acquisition and frame rate frequencies were the same as described in the reviewer comments to Fig. 1: We imaged whole microvessels (avg 20-25 μm in depth) with high z-resolution (z-step: 2.5 μm) at high speed to achieve a frame rate of 20 seconds. Laser power was adjusted to a minimum to avoid phototoxicity and photobleaching (30% of max). Platelet motility pattern were quantified using the same criteria as described in our response to the reviewers' comments on Fig. 1: while migratory platelets show a displacement larger than one cell diameter, stationary platelets do not move. Rolling platelets only transiently adhered and therefore were only visible for less than 3 frames. We now performed additional adoptive transfer experiments (n=3 mice) and extended the analysis of motility pattern of transfused wildtype and Arpc2 $-/-$ platelets as suggested by the reviewer (see Fig. 4c).

Figure 4:

- In vivo experiments were stated to be performed with adjusted platelet count between WT and Ko platelets. Platelet/visual field are less in Ko mice than in WT (Fig 4a, e, f). Can you comment on this? 4f: this would also explain the reduced colocalization of platelets to low alpha-SMA areas.

Thank you very much for this comment. As outlined in Supp. Fig 5i, we observed similar recruitment of platelets to the microvasculature in Arpc2 KO mice. We now included example images that are more representative of the quantified recruitment patterns (Fig 4 a,e, and f).

- Platelet distance from endothelial junctions is reduced in KO mice. Are vessel size, vessel number, vessel diameter and endothelial junctions comparable between KO and WT mice? Is fibrinogen

deposition on the vessel wall comparable between KO and WT mice? Are ECM proteins exposed as well and influence platelet haptotactic behavior?

Thank you for this important remark. We performed additional experiments to evaluate vessel size, number, density and size (diameter) distribution in Arpc2 +/+ and Arpc2 -/- mice (see Reviewer 1 Figure 5 c). We did not observe significant differences. We also included data on fibrin(ogen) deposition, see Suppl. 5f, which showed similar deposition. To analyze if ECM proteins, such as collagen influence platelet haptotactic behavior, we generated substrates coated with both fibrinogen and collagen fibers. Platelets adhering to fibrinogen started to migrate while platelets getting in contact and interacting with collagen stopped migrating. Consequently, during the observation time platelets progressively associate with collagen fibers (Reviewer 1 Figure 5 d). We hypothesize that platelets may follow substrate gradients (haptotaxis) to sites of ECM exposure, where they subsequently stop and seal vascular damage. We are right now investigating this further in a follow-up project.

- Image of Arpc2c-KO mice displays an increased number of PMN in inflamed cremaster. Is this difference statistically significant or is this a non-representative image?

Thank you for noting this. We did not detect a statistically significant difference in PMN recruitment to the inflamed cremaster of Arpc2-/- platelets compared to control (Suppl. Fig. 5h). We apologize for showing a non-representative image and changed it in Fig. 4 of the revised version of the manuscript.

- 4g: How were the image analyzed for microbleedings? Mice were perfusion fixated to remove all intravascular RBCs. Did the authors do a complete stack of the cremaster muscle and a 3D reconstruction of all vessels to make sure that residual Ter119 staining is outside the vessel? This is not clear from the images. Does *n* represent images or animals?

Thank you for this comment. Mice were perfusion fixated. However, in some vessels, some intravascular RBCs remained. Therefore, CD31 stained vessels were outlined and RBC signal only outside of vessels was quantified as “bleeding spots”. We now further clarified this aspect of analysis in the Methods section. The stated *n*-number represents animals.

Figure 5:

- Is fibrinogen deposition altered in the lung of KO mice?

Thank you for proposing this control experiment. We performed additional experiments showing no difference in fibrin(ogen) deposition in Arpc2 -/- compared to Arpc2 +/+ mice (see Reviewer 1 Figure 6 a).

- Is the amount of PMN-platelet conjugates comparable in KO and WT?

Thank you for proposing this control experiment. We performed additional experiments showing no difference in PMN- and monocyte-platelet-aggregate formation in Arpc2 -/- compared to Arpc2 +/+ mice (see Reviewer 1 Figure 6 b).

- Sup Fig 5g, h: Please provide platelet counts of mice after anti-hIL4-antibody mediated platelet depletion before and after the transfer of WT/KO platelets.

Thank you for this comment. We performed adoptive transfer experiments according to a previously published protocol from the Bergmeier lab (see Boulaftali et al, *JCI*, 2013). Due to a large amount of blood necessary for assessing absolute platelet counts in a hem analyzer, risking the survival of the mice in the LPS-induced ALI model, we did not perform those measurements. Instead, we controlled efficiency of depletion by measuring hIL4R-pos-platelets using flow cytometry. We observed a complete depletion of hIL4-pos platelets after anti-hIL4-antibody injection, while the relative fraction of transfused hIL4-neg platelets increased to 100% (see Reviewer 1 Figure 6c). Transfusion efficiency was around 8% of baseline platelet counts (in both WT and KO) indicating that transfused mice remain thrombocytopenic, which is also reflected in the increased bleeding tendency of WT-transfused mice compared to untreated WT animals (see Reviewer 1 Figure 6d and Suppl. Fig 6 i-k).

- 5d and Sup. Fig 5m: Values for Hgb in Bal more than 10 fold lower than in other Hgb measurements, but erythrocyte count in BAL is comparable. Do the authors have any explanation for this?

We thank the reviewer for this important remark. Indeed, the mean Hgb of *Arpc2*^{-/-} in Fig. 5d is ~0.05 g/dl corresponding to a mean of 3×10^6 erythrocytes/ μ l detected in BAL (Sup. Fig. 6c), while the mean Hgb of *allb*^{-/-} (in Suppl Fig. 6m of the revised manuscript) is ~0.015 g/dl corresponding to a mean of 4.5×10^5 erythrocytes/ μ l detected in BAL. Consequently, erythrocytes/ μ l in BAL of *Arpc2*^{-/-} was 6-fold higher compared to erythrocytes/ μ l in BAL of *allb*^{-/-}, while Hgb was only 3-fold higher. While absorbance-based Hgb measurements reflect total bleeding, TER119 based flow cytometric assessment only detects intact erythrocytes. We cannot exclude that some erythrocytes get damaged during extravasation and are no longer detectable in flow cytometry. This could be more severe in *allb*^{-/-} mice, which were bone chimera that likely have an altered/fibrotic lung microenvironment, explaining the discrepancy and non-linearity of the increase of Hgb and erythrocytes/ μ l when compared to non-chimeric mice. Therefore and since we are aware, that genetic background, age, and sex of mice have a major impact on the severity of inflammatory bleedings, we always performed appropriate controls for each set of experiments.

- Can you explain the different results in inflammatory lung bleeding after LPS treatment regarding a pharmacological inhibition of *allb* and genetic deletion?

We assessed lung bleeding in an established LPS induced inflammatory bleeding model over a time course of 24 hours (see Benoit Ho-Tinh-Noe et al, *Haematologica* 2018). Most pharmacological inhibitors like tirofiban have only a short plasma half-life (around 2 hours, Kumar et al, *Expert opinion on investigational drugs*, 1997) and would therefore have to be injected multiple times during the course of the experiment. Consequently, transient pharmacological inhibition (after a single dose of tirofiban at the beginning of the experiment) is unlikely to completely block integrin $\alpha_{IIb}\beta_3$ over the entire experiment. Indeed, we were able to reproduce this data and did not detect inflammatory bleeding after 24h of ALI following a single dose of Tirofiban (see Reviewer 1 Figure 6e). To overcome this

methodological problem, we therefore chose genetic ablation of integrin $\alpha_{IIb}\beta_3$ as the state-of-the-art method to discern the role of $\alpha_{IIb}\beta_3$ in our ALI model.

Figure 6:

- Movie 6: are MRSA bacteria engulfed in platelets or do they bundle on to the surface?

Platelets are bundled on the surface of platelets (see Gaertner et al, *Cell*, 2017). We included a scanning electron microscopy image to highlight this fact (see Fig 7c).

- 6a: What does PH stand for?

We apologize for this; we clarified that PH stands for phase contrast in Fig 7a of the revised manuscript.

- 6e: How biologically relevant is reduction from 3% to 1% of MRSA associated with platelets? At which time point after inoculation were d and e analyzed?

We thank the reviewer for this comment. Fig. 6d and e were analyzed at 48 hours post infection. After 48h of infection the large majority of bacteria accumulated in abscesses surrounded by PMN. A smaller fraction of bacteria was localized outside of abscesses and we hypothesized that those bacteria are most likely to get access to the blood stream and disseminate within the host. We show that single platelets are recruited to the weakened inflammatory microcirculation of the lung where they maintain vascular integrity. Consequently, platelets also occupy strategic positions at the vascular border where they can immediately bind to bacteria that start to invade the microvessels. We show that 3% of all bacteria outside of abscesses are bound by platelets that might prevent their entry into the bloodstream. This interaction is reduced by two-thirds in KO mice (1%). Based on these observations we hypothesized that platelet-specific *Arpc2*-deficiency might exacerbate bacteremia in our lung infection model. Indeed, both the fraction of bacteremic mice as well as CFUs measured in the blood were largely increased in *PF4Cre+;Arpc2^{fl/fl}* mice compared to control, highlighting the biological relevance of our findings. Since we did not observe other major histological differences between control and KO mice, such as numbers of recruited platelets and PMNs or abscess formation (Supp Fig 7 f-k), the reduction in platelet-bacteria interactions is likely to be causative in increased bacteria dissemination in blood and organs.

Minor comments:

- Fig 1b: How many mice were analyzed? Figure legends just indicate 39 platelets. Please check complete manuscript and discriminate between number of experiments and number of mice.

Thank you for this comment. We added this information where applicable and analyzed at least three mice in every experiment.

- 3j-l: WT mice were treated with busulfan to reduce the plt count to KO-levels. Please include this in the figure legend and figure labelling.

We included this data.

- Length of scale bare missing in figure legend 1 for all microscopic images.

We included this data.

- In general, due to the overload of data, explanations are kept too short or too general- especially regarding the supplemental figures (e.g. no explanation of Blebbistatin and cytochalasin treatment in supplemental Figure 4).

Thank you, we added an extended description of this experiment in the legend and methods section.

Point-to-point response to Reviewer #2:

We thank the reviewer for the positive evaluation of our manuscript and for her/his valuable and thoughtful comments. Please find below our point-to-point response.

Reviewer #2 (Remarks to the Author):

In the present manuscript 'Vascular surveillance by haptotactic blood platelets in inflammation and infection', Nicolai and colleagues report a critical role of Arp2/3 complex-mediated lamellipodia formation in the sealing of vascular microinjuries post neutrophil extravasation (assessed upon LPS-mediated cremaster and lung inflammation) in a fibrinogen-dependent manner. In addition, the authors show that platelets prevent bacteria and erythrocyte dissemination in the context of infection. With this the authors report for the first time a role of migrating, haptotactic platelets in safeguarding inflamed vessels. Although the study is novel, extensive, well designed and controlled as well as of great interest to the broad readership of Nature Communications, there are some aspects that should be addressed.

Major comments:

1. How do the authors explain the absence of microbleeds/inflammation-associated bleeding in the lung of other mouse models incapable of forming platelet lamellipodia, e.g. *Cyfp1*^{-/-} (Schurr et al. Blood 2019). Similarly, why does the reverse passive Arthus reaction not cause bleeding in the skin of *Arpc2*^{-/-} (Paul Blood Adv 2018) and *Cyfp1*^{-/-} (Schurr et al. Blood 2019)? This should at least be discussed.

To address this very important question we performed additional experiments with *PF4*-Cre+; *Cyfp1*-flox mice (data shown in Figure 6 of the revised version of the manuscript) and added an additional paragraph to the results section explaining why *PF4*-Cre+; *Cyfp1*-flox mice did not show microbleeds/inflammation-associated bleeding in the lung:

"WASp and WAVE play a redundant role in platelet haptotaxis

*The Arp2/3 complex of platelets is activated by nucleation-promoting factors (NPFs) of the Wiskott-Aldrich Syndrome protein (WASp) and WASp-family verprolin-homologous protein (WAVE) families^{16,39}. To test the contribution of these NPFs in platelet migration and haptotaxis we analyzed platelets deficient in WASp or Cyfp1, the Rac1-binding subunit of the WAVE complex required for its activation in platelets²³. WASp^{-/-} platelets formed lamellipodia similar to control platelets and showed unimpaired migration and fibrinogen removal, indicating a redundant role of WASp in lamellipodia-formation and platelet migration (Fig. 6a). Previous reports have shown that WAVE rather than WASp is required for lamellipodia formation in platelets^{23,40}. Indeed, spreading area of platelets isolated from conditional *Pf4cre* +/-; *Cyfp1*^{fl/fl} (*Cyfp1*^{-/-}) mice was largely reduced compared to controls (Fig. 6b) and adherent platelets formed prominent filopodia (Fig. 6b), confirming previous observations²³. By using time-lapse video-microscopy we next analyzed the dynamics of protrusions of platelet from *Cyfp1*^{-/-} mice. Interestingly, *Cyfp1*^{-/-} platelets form premature lamellipodia-like extensions that protrude and retract between adjacent filopodia, but do not extend to filopodial tips (Suppl. Video 6) leading to projected*

platelet areas that were slightly, but significantly larger compared to *Arpc2*^{-/-} platelets (Fig. 6c-d). *Cyfp1*^{-/-} platelets showed a very mild, but not significant reduction of migration and unimpaired removal of fibrinogen, indicating that premature lamellipodia-like extensions were sufficient to promote platelet migration (Fig. 6e-g). Incubation of *Cyfp1*^{-/-} platelets with the *Arp2/3* inhibitor CK666 abolished migration, indicating residual *Arp2/3* activation of *Cyfp1*^{-/-} platelets sufficient to promote migration (Fig. 6h). Hence, *Cyfp1*^{-/-} platelets were able to haptotax and preferentially migrated to higher fibrinogen densities (Fig. 6i). During ARDS, *Cyfp1*^{-/-} platelets prevented inflammatory bleedings of the lung (Fig. 6j), despite reduction of platelet counts to the level of *PF4-Cre;Arpc2*^{fl/fl} mice (Fig. 6k), indicating functional haptotaxis *in vivo*. Consequently, and similar to previous reports of other cell types⁴¹, our data suggests a redundancy of NPFs in the activation of the *Arp2/3* complex, which is sufficient to promote migration when one or the other is deleted.”

In addition, we also added a paragraph to the discussion:

“*Arp2/3* in platelets is activated by nucleation-promoting factors (NPFs) of the WAVE and WASp families, as well as cortactin and HS1^{23,40,62}. Patients with loss-of-function mutations of WASp present with immunodeficiency and bleeding complications, known as Wiskott-Aldrich-Syndrome⁶³. Consequently, it was tempting to speculate whether impaired platelet migration and haptotaxis contribute to the clinical symptoms of WAS patients. Our data shows that single deletion of neither WASp nor WAVE was sufficient to fully inhibit *Arp2/3* activation and to block platelet migration and haptotaxis, suggesting redundancy of NPFs in this process⁴¹. In contrast to WASp^{-/-}, WAVE deletion (*Cyfp1*^{-/-}) abrogated large circular lamellipodia in platelets²³. However, the short *Arp2/3*-dependent lamellipodial-like extensions forming at the center of *Cyfp1*^{-/-} platelets, were still sufficient to promote platelet migration and haptotaxis, indicating a highly efficient molecular machinery supporting platelet motility. Further studies will be required to shed light on the mechanisms of *Arp2/3* activation in migrating platelets and to investigate the mechanistic framework underlying the redundancy of NPFs.”

2. Due to the emerging evidence from single cell RNA sequencing experiments on different origins, molecular identities and functions of endothelial cells from different vascular beds would it be of value to additionally visualize platelet migration in another easily accessible vascular bed, e.g. mesentery/ear skin to exclude vascular bed-specific effects. Are the observed effects specific to LPS-mediated inflammation? What happens in response to a more generalized inflammation, e.g. in response to TPA? Moreover, proof of vascular integrity breakdown is missing, injection of fluorescent beads could help to demonstrate vascular leakage. This is specifically evident in Fig. 1c-d. In 1c junctions are clearly discontinuous, which is in agreement with vascular leak but not in 1d, where junctions still show a zipper-like structure.

We thank the reviewer for this important comment. We were able to show that *Arp2/3*-dependent platelet migration and haptotaxis is important in preventing lung and muscle bleeds after LPS challenges, therefore highlighting crucial importance of this novel effector function in at least two vascular beds. To address if this platelet function is stimulus-dependent, we established and performed an additional inflammation model using CCL-2 as a model-stimulus of sterile inflammation in the cremaster. We included this novel data in the revised text and figures (see Suppl. Fig 5k). Interestingly, *Arpc2*^{-/-} mice showed increased bleeding also in CCL-2-mediated inflammation. This model showed an

even more striking difference in bleeding compared to LPS stimulation, which corresponds well to the increased extravasation of neutrophils observed in this model (please see Zuchriegel et al, *PLoS Biology*, 2017). In conclusion, we show that platelet migration and haptotaxis is relevant in different vascular beds (lung and muscle) independent of inflammatory stimulus (sterile inflammation: CCL-2 and bacterial-TLR4 sensing: LPS). We also tested a model of bead extravasation, as proposed by the reviewer, using beads similar to Hillgruber et al, *JEM* 2015. We used platelet depletion by R300, known to cause extensive bleeding (see Suppl. Fig 5 a-e). As expected, we saw significantly increased erythrocyte extravasation in depleted animals (Reviewer 2 Figure 1 a); however, bead accumulation showed no increase in R300 depleted mice, and did not correlate with TER119 (erythrocyte) signal (Reviewer 2 Figure 1 b). Interestingly, we observed “stuck” beads within the inflamed vasculature (Reviewer 2 Figure 1 a), and co-staining of Lyg6+ neutrophils revealed uptake of beads by intravascular neutrophils (Reviewer 2 Figure 1 c). We therefore conclude that in our hands, using beads did not confer advantage over an optimized extravascular erythrocyte staining. The reason for this could be that we studied microbleeds as opposed to macroscopic bleeding, and that our assay over 6 hours is not suitable for bead-based studies due to rapid clearance of beads from the circulation by macrophages and neutrophils. Also, the uptake of beads by neutrophils makes this assay more prone to bias, as areas of substantial neutrophil recruitment will accumulate beads independently of vascular breakdown.

3. The authors clearly show that platelet morphology and the underlying mechanisms differ between inflammation and thrombosis. This raises the question on whether platelet migration plays a role in thrombo-inflammatory processes post transient mid cerebral artery occlusion.

Thank you very much for this valuable remark. Indeed the tMCAO mouse model would be of particular interest, as it has been shown to depend on $\alpha_{IIb}\beta_3$ function. While these experiments go beyond the scope of this study it will be interesting to address this question in follow-up projects.

In addition, what happens upon induction of minor endothelial damages e.g. by using less FeCl₃ (e.g. 7%) and/or reduced incubation time which does not lead to full vessel occlusion?

Thank you for this comment. *Paul et al. (Blood advances 2017)* used a complete FeCl₃ occlusion model and did not observe differences in thrombus formation in *Arpc2* *+/+* and *-/-* mice. In previous studies of our group, we observed that increased FeCl₃ injury leaves no room to detect minor changes of platelet function due to the extensive damage (Pircher et al, *Nat Comm* 2017, Marx et al, *Blood* 2019). In this study we therefore decided to use a milder injury with only partial occlusion (also see Methods). However, even by using this milder injury with only partial vessel occlusion, we were able to confirm the data of *Paul et al.* and did not observe a significant difference in thrombus formation between KO and WT mice (see Suppl. Fig. 4 a). In addition, these findings are also in line with a recent study by *Schurr et al (Blood 2019)*, highlighting that lamellipodia are dispensable for thrombus formation *in vivo*.

4. Nicolai and colleagues convincingly demonstrate that platelets migrate towards higher ligand densities. However, based on the tracks presented in Fig. 3e it is tempting to speculate that additional mechanisms might contribute to platelet ‘decision’ making. How do the author’s explain that all

displayed platelets on the low density matrix (even the ones farther away) migrate towards the high density fibrinogen interface although they are surrounded by a homogenous fibrinogen matrix and thus should display a random and not directed migration?

Thank you very much for this comment. We are sorry for not explaining the setup of this experiment in a more comprehensive way. We only included cells in our analysis that came into contact with the interface of low-and-high fibrinogen density. Migrating cells that did not get in contact with this interface were **not** included in the analysis. Indeed, platelets not coming into contact with the low-to-high fibrinogen gradient showed no propensity to migrate towards the higher concentration, highlighting that platelets are haptotactic (See Reviewer 2 Figure 1 d). We clarified this aspect in the text and figure legend of the revised manuscript.

5. Overall it is difficult to follow in which mouse model ('native' control and mutant *Arcpc2* mice, platelet count-adjusted (antibody- or drug-mediated), bone marrow chimeras or platelet transfusion) the presented experiments were performed.

Thank you very much for this comment. We now included a table stating the used mice and experimental setups (see Supplementary table 1).

Additional information on the model and explanation on why the model was chosen would help to improve the comprehensibility of the study.

Thank you very much for the comment. We include additional information for each model and explanation for the selection of each model in the table setups (see Supplementary table 1).

Depending on the used model, certain controls are missing, e.g. fibrinogen localization in *Arcpc2*^{-/-} mutants. Unaltered fibrinogen distribution would strengthen the notion of impaired migration.

Thank you very much for this comment. We now compared fibrinogen deposition *Arcpc2*^{-/-} and control mice, please refer to Suppl. Fig 5f and Reviewer 1 Fig. 6a.

Moreover, *Arcpc2*^{-/-} pictures in Figure 4a, e, f do not reflect statistics presented in Figure 4c and Supplementary Figure 4h as all of them show less platelets for mutants than controls.

Thank you very much for this comment. As outlined in Supp. Fig 4f, we observed similar recruitment of platelets to the inflamed microvasculature in both *Arcpc2* KO mice and WT controls. We now included example images more representative for the quantified recruitment patterns (Fig 4 a, e, f).

Why is the adhesion of transfused but not native *Arcpc2*^{-/-} platelets to inflamed cremaster vessel almost doubled (Fig. 4c; Supplementary Fig. 4h)?

Thank you very much for this comment. We apologize for not properly explaining our setup. We transfused equal counts of Arpc2 $-/-$ and Arpc2 $+/+$ platelets labeled in different colors into wild type mice, and elicited cremaster inflammation by injecting LPS. We then imaged the cremaster microvasculature and compared platelet motility pattern of transfused WT and KO platelets (see Fig 4c for illustration and results). We also apologize for not being precise in the naming of motility patterns, which we now corrected in the revised manuscript. “Adhesion” refers to adhesive cells that are stationary and do not migrate (now referred to as “Stationary”). “Migration” refers to adhesive cells that are motile and do migrate. Consequently, not the fraction of cells adherent to the vascular wall (which is the sum of stationary plus migratory cells) but rather the fraction of stationary cells is almost doubled.

Is integrin trafficking also altered/ contributing to the observed defects? It would be interesting to assess integrin turnover and closure which were previously shown to be dependent on remodeling of the cortical actin cytoskeleton.

We do not think that integrin turnover is severely affected, as Arpc2 $-/-$ platelets and CK666-inhibited platelets showed unimpaired $\alpha_{IIb}\beta_3$ -dependent clot-retraction, a platelet function critically dependent on integrin-activation (see Suppl. Fig 3b,g, j). In addition, activation of $\alpha_{IIb}\beta_3$ as measured by FACS was not changed upon stimulation with various activators (see Suppl. Fig 3 i) and baseline levels of $\alpha_{IIb}\beta_3$ expression was comparable (see Suppl. Fig 3 i).

6. The presented pictures on the acute lung injury model (Fig. 5a,b,e) are not very informative. inclusion of Cdh5/Cd31 staining would help to obtain some morphological information. It's difficult to see lamellipodia in the presented pictures. Pictures of untreated controls for comparison are missing. How did the authors distinguish between intact vasculature and bleeding spots for their quantifications? Again, beads could help to address this question.

Thank you very much for this comment. We performed additional experiments and staining to show platelet spreading and bleeding in relation to the lung vasculature (CD31 staining). This data has been included in Fig. 5 a and b of the revised manuscript. We quantified bleeding by perfusing the mice and then identifying areas of increased TER119 signal that were quantified as “bleeding spots”. We confirmed that these erythrocytes were outside the vessel (see Fig. 5 b). We also included now examples of NaCl treated mice for comparison (Fig 5 a and b). In addition, we completed additional experiments, where we injected fluorescently labeled beads and then compared extravasation in NaCl and LPS treated animals. Similar to our findings in the inflamed Cremaster muscle, beads did not correlate with increased TER119 signal in LPS treated animals (see Reviewer 2 Figure 1 e-f). We therefore did not include this data in the main manuscript, but would be happy to do so, if the reviewer deems this necessary.

7. Fig 6e: Why was this model performed in bone marrow chimeras? Were platelet counts equal for both groups?

To generate a large number of sex and age-matched knock-out and control mice required for the infection experiments, we generated bone marrow chimeras to overcome limited breeding capacities.

To compensate for the thrombocytopenia of *Arpc2*^{-/-} chimeras, platelet counts of *WT*-chimeras were adjusted using Busulfan depletion (also see Suppl. 4 d-f). Consequently, platelet recruitment into infected lungs of *WT* and *Arpc2*^{-/-} chimeras was not significantly different (see Suppl. Fig 7h).

Minor comments:

1. Line 104: “Supplementary Fig. b-d” should read as Supplementary Fig. 1b-d.

Thank you, we corrected this.

2. Line 250: “Supplementary Fig. 7h” should read as Supplementary Fig. 3h.

Thank you, we corrected this.

3. Fig 4h and Suppl. Fig. 4e, f: Why are there so many intravascular erythrocytes despite perfusion?

Thank you for this comment. We did perform perfusion prior to cremaster preparation in these experimental setups. Sometimes, intravascular erythrocytes remained in the vessel. We cannot definitely pin this down to a specific reason; possibly, intravascular inflammation associated thrombosis might prevent proper transfusion of singular vessels. We circumvented this problem by only quantifying extravascular TER119 (Erythrocyte) signal by removing all Erythrocytes localized with CD31+ vessels from bleeding analysis.

4. The statement ‘Taken together, Arp2/3-dependent spreading and haptotaxis constitute crucial platelet functions required for safeguarding the inflamed microvasculature and preventing inflammatory bleeds.’ seems slightly overstated considering the unaltered results of the rpA model in *Arpc2*^{-/-} animals (Paul Blood Adv 2018).

Thank you very much for this comment. In this study we provide first evidence of platelet haptotaxis as a novel immune-related platelet function and we agree that further studies using different inflammation models are required to fully understand the role of platelet haptotaxis in preventing inflammatory microbleeds. We therefore relativized our statement in the revised manuscript as follows: ‘Taken together, Arp2/3-dependent spreading and haptotaxis constitute crucial platelet functions required for safeguarding the inflamed microvasculature of muscle and lung and preventing inflammatory microbleeds.’

5. Fig. 6a: Is the biofilm clearance already correlated to the number of platelets or does it just reflect the increased number of recruited platelets and hence clearance in the presence of MRSA?

Thank you very much for this question. Indeed, the data presented in 6a is not corrected for number of recruited platelets. It therefore highlights the ability of MRSA to activate and recruit platelets to biofilms, therefore triggering clearance thereof; in contrast, cleared area per cell – included in Reviewer 2 Figure 3g - did not differ between biofilms with and without MRSA .

Point-to-point response to Reviewer #3:

We thank the reviewer for the positive comments and appreciation of our work. Please find below our point-to-point response.

Reviewer #3 (Remarks to the Author):

The authors examine the role of Arp2/3 complex in regulating platelet shape and motility. They use an impressive array of in vitro and in vivo approaches in their study. Their findings reveal an important role for Arp2/3 in positioning platelets to microlesions in the inflamed microvasculature and capture of pathogenic bacteria. This is a superb paper that sheds important light on platelet movement in vivo and the role of Arp2/3 in general. I have only minor comments for the authors consideration.

-Page 5. “we detected a thin layer of fibrin(ogen), providing a two-dimensional platform for adhesion of single platelets in vivo (Fig. 1c).”. Can the authors comment further on the pattern of fibrin(ogen) deposition observed? Actual depth? Patterns? Structures?

Thank you very much for this comment. The distribution of fibrin(ogen) within inflamed microvessels, indeed follows a specific pattern. We identified fibrin(ogen) deposition is largely restricted to endothelial junctions with density gradients peaking at the center of junctions (see Fig 4 d). To measure the actual depth of fibrin(ogen) deposits we performed 3D-reconstruction of wholemount super-resolution microscopy data using IMARIS software. While z-resolution was sufficient to demonstrate that platelets were spreading on top of and not underneath thin deposited fibrin(ogen) layers (see Figure 1 c inset) the z-resolution of this technique does not allow to measure the precise thickness of these deposits. However, given the fact that the reconstructed fibrin(ogen) deposits have a thickness comparable to adherent platelets one can estimate an actual depth around <1 μm.

-Page 6 “Importantly, cleavage of 3D fibrillar fibrin mesh-works by the serine-protease plasmin rescued polarization and migration, highlighting the ability of platelets to instantaneously adapt their motility program to the physical composition of the microenvironment (Fig. 2e-f and Supplementary Video 2).”. These are interesting data but may reflect other effects of the enzyme. Have the authors tried an inactive (e.g. denatured) enzyme control?

Thank you very much, we performed an additional experiment with heat-inactivated plasmin, which failed to degrade fibrin fibers and change platelet behavior (see Reviewer 3 Figure 1 a).

-Figure 6g, The authors report CFUs/mL blood. The y-axis should be Log10 scale.

Thank you very much, we included log-scale data.

-Figure 6h. CFU data should be included for kidney and spleen for comparison with g.

Thank you. We now included CFU data in Suppl. Fig 7.

-The authors may wish to discuss Wiscott Aldrich Syndrome, which pointed to Arp2/3 regulation in immune control.

We thank the author for this very important suggestion. To address this comment, we performed additional experiments (Figure 6 of the revised manuscript) and add an additional paragraph to results:

“WASp and WAVE play a redundant role in platelet haptotaxis

The Arp2/3 complex of platelets is activated by nucleation-promoting factors (NPFs) of the Wiskott-Aldrich Syndrome protein (WASp) and WASp-family verprolin-homologous protein (WAVE) families^{16,39}. To test the contribution of these NPFs in platelet migration and haptotaxis we analyzed platelets deficient in WASp or Cyfip1, the Rac1-binding subunit of the WAVE complex required for its activation in platelets²³. WASp^{-/-} platelets formed lamellipodia similar to control platelets and showed unimpaired migration and fibrinogen removal, indicating a redundant role of WASp in lamellipodia-formation and platelet migration (Fig. 6a). Previous reports have shown that WAVE rather than WASp is required for lamellipodia formation in platelets^{23,40}. Indeed, spreading area of platelets isolated from conditional Pf4cre +/-; Cyfip1fl/fl (Cyfip1^{-/-}) mice was largely reduced compared to controls (Fig. 6b) and adherent platelets formed prominent filopodia (Fig. 6b), confirming previous observations²³. By using time-lapse video-microscopy we next analyzed the dynamics of protrusions of platelet from Cyfip1^{-/-} mice. Interestingly, Cyfip1^{-/-} platelets form premature lamellipodia-like extensions that protrude and retract between adjacent filopodia, but do not extend to filopodial tips (Suppl. Video 6) leading to projected platelet areas that were slightly, but significantly larger compared to Arpc2^{-/-} platelets (Fig. 6c-d). Cyfip1^{-/-} platelets showed a very mild, but not significant reduction of migration and unimpaired removal of fibrinogen, indicating that premature lamellipodia-like extensions were sufficient to promote platelet migration (Fig. 6e-g). Incubation of Cyfip1^{-/-} platelets with the Arp2/3 inhibitor CK666 abolished migration, indicating residual Arp2/3 activation of Cyfip1^{-/-} platelets sufficient to promote migration (Fig 6h). Hence, Cyfip1^{-/-} platelets were able to haptotax and preferentially migrated to higher fibrinogen densities (Fig. 6i). During ARDS, Cyfip1^{-/-} platelets prevented inflammatory bleedings of the lung (Fig. 6j), despite reduction of platelet counts to the level of PF4-Cre;Arpc2^{fl/fl} mice (Fig. 6k), indicating functional haptotaxis in vivo. Consequently, and similar to previous reports of other cell types⁴¹, our data suggests a redundancy of NPFs in the activation of the Arp2/3 complex, which is sufficient to promote migration when one or the other is deleted.”

In addition, we also added a paragraph to the discussion:

“Arp2/3 in platelets is activated by nucleation-promoting factors (NPFs) of the WAVE and WASp families, as well as cortactin and HS1^{23,40,62}. Patients with loss-of-function mutations of WASp present with immunodeficiency and bleeding complications, known as Wiskott-Aldrich-Syndrome⁶³. Consequently, it was tempting to speculate whether impaired platelet migration and haptotaxis contribute to the clinical symptoms of WAS patients. Our data shows that single deletion of neither WASp nor WAVE was

sufficient to fully inhibit Arp2/3 activation and to block platelet migration and haptotaxis, suggesting redundancy of NPFs in this process⁴¹. In contrast to WASp^{-/-}, WAVE deletion (Cyfip1^{-/-}) abrogated large circular lamellipodia in platelets²³. However, the short Arp2/3-dependent lamellipodial-like extensions forming at the center of Cyfip1^{-/-} platelets, were still sufficient to promote platelet migration and haptotaxis, indicating a highly efficient molecular machinery supporting platelet motility. Further studies will be required to shed light on the mechanisms of Arp2/3 activation in migrating platelets and to investigate the mechanistic framework underlying the redundancy of NPFs.”

a

b

c

d

e

f

g

a

b

c

d

e

Fbg AF488 1:20 Fbg Ab AF647 MERGE

Line profile

		Concentration		Assay used volumes		Fbg	Proportion labeled
Plasma Fbg	300 mg/dl	3 mg/ 1ml	3 μg/ μl	60 μl	180 μg	1:20	
Fbg488		1,5 mg/ml	1,5 μg/ μl	6 μl	9 μg		
		Concentration		Blood total			
Plasma mouse		1,5 mg/ml	1,5 μg/ μl	2 ml	3000 μg	1:17	
Fbg488 inj.		1,5 mg/ml	1,5 μg/ μl	120 μl	180 μg		

f

Reviewer 1 Figure 3

a

b

c

d

e

Reviewer 1 Figure 4

a

b

c

d

e

Reviewer 1 Figure 5

a

b

c

d

Reviewer 1 Figure 6

Reviewer 2 Figure 1

Reviewer 3 Figure 1

a

Reviewer Figure Legends:

Reviewer 1 Fig. 1: Mouse-based analysis. **a**, Cells per motility pattern, directionality, velocity and accumulated distance, Platelets recruited to inflamed blood vessels are migratory and form lamellipodia, see Fig 1 and Suppl. Fig1. n=5 mice, **b**, mouse based analysis of data in Fig 4., n=3 mice **c**, mouse based analysis of shape descriptors in inflammation and thrombosis, see Fig 1. n=3 mice., **d**, platelet retraction and migration velocities *in vivo*, see Fig 1, n=3 mice, **e**, clot retraction *in vivo* comparing Arpc2 +/+ and -/- mice, see Suppl. Fig 4. n=3 mice. **f**, mouse based analysis of mean platelet distance from endothelial junctions, and % of alpha-SMA depleted extravasation sites enriched in platelets, mouse based analysis, n=3 mice, compare Fig. 4. **g**, mouse based-analysis of motility patterns of transfused Arpc2 -/- and +/+ platelets in the inflamed cremaster microvasculature, compare Fig. 4, n=3.

Reviewer 1 Fig. 2: **a**, motility patterns of transfused Myh9+/+ and -/- platelets. Quantification of patterns (n=16 vessels from 2 mice), Mann-Whitney The experiment was performed with the transfusion of PF4-cre + and - Myh9 fl/fl platelets (see Gaertner et al, *Cell*, 2017) in a setup similar to the one used in Fig 4c.b, *in vivo* mesenteric clot retraction in Myh9 -/- and +/+ mice (compare Suppl. Fig 4). n=48-73 platelets, 2 mice per group. **c**, recording time of 4d spinning disc videos, see **Fig 1**. **d**, LPS induced cremaster inflammation model compared to NaCl control, left: exemplary micrographs showing platelet, PMN, and fibrinogen recruitment/deposition in NaCl/LPS treated mice. right: quantification of platelet, PMN, and fibrinogen recruitment/deposition, n=3 mice per group, ttest, scale bar: 20 μ m. **e**, *in vitro* cross-linked fibrin substrate spiked with 1:20 AlexaFluor488 labeled fibrinogen, stained with Anti-Fibrinogen Ab and secondary AlexaFluor 647 antibody. Fluorescence distribution in x-y and z-axis. Table: Aproximation of injected labeled fibrinogen compared to the *in vitro* cross-linked fibrin substrate spiked with labeled fibrinogen. **f**, analysis of *in vivo* platelet shape in the cremaster microvasculature, n=26 (thrombosis), n=33 (inflammation) from 3 mice, see **Fig 1h** for reference. 54.54% of deposited cells showed larger area than platelets recruited to the thrombi, indicating spreaded morphology.

Reviewer 1 Fig. 3: **a**, no evidence for lamellipod formation in thrombosis. Single platelet morphology in the early phase of mesenteric thrombus formation (see methods: mesenteric clot retraction model). star: filopodium of adherent platelet. Right: quantification of cell area in the thrombus, and circularity of recruited cells. Scale bar: 5 μ m. **b**, human, baboon and mouse platelets seeded on fibrinogen or cross-linked fibrin surfaces, revealing similar responses to substrates, with migration occurring on Fibrinogen, and retraction occurring on cross-linked fibrin substrate. **c**, WT mouse platelet migration is comparable to isotype in the presence of GPVI blocking Jaq1 antibody. Scale bars: 5 μ m. **d**, plasmin mediated cleavage of cross-linked fibrin substrates in the absence of platelets, leading to removal of fibers and loss of fluorescence signal. Scale bar= 5 μ m. ttest. **e**, heat-inactivated plasmin has no effect on platelet behavior and does not lead to fibrin fiber cleavage. Methodology: see methods section.

Reviewer 1 Fig. 4: **a**, Western Blot analysis of platelet lysate of PF4-cre + and PF4-cre - Arpc2 fl/fl showing loss of Arp2/3 complex in Arpc2 deficient platelets. **b-e**, analysis of platelet function in WT mice partly platelet depleted by busulfan and control mice (see methods), day=14, (b), activation flow cytometric measuring JonA binding, n=4 mice (c), clot retraction over time and as % of volume, n=4 mice, (d), platelet shape analysis on fibrinogen substrates, (left) representative micrograph, (right) cell area, solidity, and circularity in Ctrl and Busulfan platelets. n=33-34 cells from 4 mice. (e), platelet migration of Ctrl and Busulfan platelets, representative micrograph, scale bar = 1 μ m.

Reviewer 1 Fig. 5: **a**, Ferric Chloride induced thrombosis model in the mouse carotid artery. n=5-6 mice. **b**, histogram and exemplary thrombus formation in the mesenteric clot retraction model (see Fig 1g and Suppl. Fig 4). Red boxes: Time/Areas of interest, where early thrombus formation enables the observation of clot retraction on a single cell level. **c**, vessel size, mean diameter and vessel area in the cremaster of Arpc2 +/+ and Arpc2 -/- mice, ttests, right: representative micrographs, vessels stained with CD31-AF647. **d**, platelet migration assay, compare Suppl. Fig 2, with addition of collagen fibers (red arrows). Platelets stop migrating once interacting with collagen (white stars). Right: collagen bound platelet show reduced migration, and migrating platelets become increasingly associated with collagen fibers, ttests, scale bar: 10 μ m. Collagen-fibrinogen surfaces were coated with 37.5 μ g/ml AlexaFluor-conjugated Fibrinogen and 25 μ g/ml Homs Collagen and 0.2% HSA in modified Tyrode's buffer for 15 min at RT (see also Methods: 2D migration assay for platelet isolation and seeding).

Reviewer 1 Fig. 6: **a**, Fibrinogen deposition in the inflamed lungs of Arpc2 +/+ and -/- mice after 8 hours post intranasal 20 μ g LPS application, ttest. Methodology: See methods: Acute lung injury. **b**, platelet-PMN and platelet- monocyte aggregates in Arpc2 -/- and Busulfan treated Arpc2 +/+ control mice. ttest. Blood was obtained 2h post intranasal 20 μ g LPS application. Right: PLAs exemplary FACS plots. **c-d**, refers to Suppl. Fig 6 g-k, adoptive transfer experiment. **(c)** flow cytometric analysis of hIL4 + and hIL4 - platelets at baseline and after depletion/transfusion of Arpc2 -/- and control platelets in hIL4R/GP1b-tg mice, and approximation plt count in % to baseline after depletion/transfusion. n=4 mice. **e**, transient allbIII blockade with tirofiban in sub-acute 24h LPS acute lung injury model has no significant effect on inflammatory bleeding. Left: protocol, right: Hemoglobin in BAL, see Subacute lung injury model in methods section for details on methodology.

Reviewer 2 Fig. 1: **a**, isotype treated (ctrl) and platelet depleted mice (R300), and induction of LPS cremaster inflammation (see methodology), injection of 4.55×10^9 dia 1 μ m inert Fluoresbrite YG microspheres (beads) at 2hrs and 4hrs post intrascrotal LPS application. Whole mount assessment of bleeding by erythrocyte (TER119) and bead (YG) signal, left: exemplary micrographs and right: quantification of TER119 and Bead signal, n= 3 mice per group. **b**, correlation with of TER119 signal and bead deposition per slide (n= 18 slides). **c**, colocalization of platelet, neutrophils and beads in the inflamed cremaster microvasculature, showing uptake and binding of beads by neutrophils. **d**, platelet decision making upon encountering substrate gradients, see **Fig 3 d-e**. (left) Quantification of platelets coming into contact with the interface migrating from low to hi and hi to low densities. (right), platelets in contact with gradient/interface and not in contact, analyzed for position in low (grey) or high (orange) substrate densities at time 0 and after 45 minutes migration. (below): analysis of 22 cells not in contact with interface/gradient analyzed for migration direction. **e-f**, NaCl control and LPS induced sub-acute lung injury. (See **Fig 6**). Injection of injection of 4.55×10^9 dia 1 μ m inert Fluoresbrite YG microspheres (beads) at 2hrs and histological analysis at 6 hrs post i.n. LPS/NaCl intranasal application. Histological assessment of bleeding by erythrocyte (TER119) and bead (YG) signal, left: exemplary micrographs and right: quantification of Ter119 and Bead signal, n= 3 mice per group. LPS treated mice show increased TER119 signal compared to NaCl treated mice, whereas beads show similar signal in both setups. **b**, correlation with of TER119 signal and bead deposition per slide (n= 12 slides). **g**, platelet recruitment and clearance in sterile and MRSA biofilms *in vitro*, see Figure 7a, analyzed for area cleared per cell with/without MRSA. n=5, Mann Whitney. Scale bar = 10 μ m.

Reviewer 3 Fig. 1: **a**, heat-inactivated plasmin has no effect on platelet behavior and does not lead to fibrin fiber cleavage.

REVIEWERS' COMMENTS:

Reviewer #1 (Remarks to the Author):

The authors did a great job to address our comments and have answered my questions satisfactorily.

Minor comments:

- Fig S1: correct labeling of Fig S1a left images are mislabeled with NaCl and LPS, the right one needs to be labelled with LPS. Description for Fig S1a Images and quantification missing in figure legend.
- What does the asterisk in Fig 5B indicate? Please indicate in figure legend.
- Fig 6c, d: x-Axis labeling should be Arpc2-/- instead of Arpc3-/-.

Reviewer #2 (Remarks to the Author):

Great work, folks!
Best regards, Simon Stritt

Reviewer #3 (Remarks to the Author):

The authors have addressed my comments. This is a superb paper.

Point-to-point response to reviewers

Reviewer #1 (Remarks to the Author):

The authors did a great job to address our comments and have answered my questions satisfactorily.

We thank you for your invaluable comments and for the effort in reviewing our manuscript.

Minor comments:

- Fig S1: correct labeling of Fig S1a left images are mislabeled with NaCl and LPS, the right one needs to be labelled with LPS. Description for Fig S1a Images and quantification missing in figure legend.

We now corrected the labeling of Figure S1a and added a description to the figure legends.

- What does the asterisk in Fig 5B indicate? Please indicate in figure legend.

We now added a description.

- Fig 6c, d: x-Axis labeling should be Arpc2-/- instead of Arpc3-/-.

We corrected the x-axis labels accordingly.

Reviewer #2 (Remarks to the Author):

Great work, folks!

Best regards, Simon Stritt

We thank you for your invaluable comments and for the effort in reviewing our manuscript.

Reviewer #3 (Remarks to the Author):

The authors have addressed my comments. This is a superb paper.

We thank you for your invaluable comments and for the effort in reviewing our manuscript.